# Surveying of Nearshore Bathymetry Using UAVs Video Stitching

**Jinchang Fan** [1], **Hailong Pei** [1,2,*] and **Zengjie Lian** [1]

1    Key Laboratory of Autonomous Systems and Networked Control, Ministry of Education,
     Unmanned Aerial Vehicle Systems Engineering Technology Research Center of Guangdong,
     South China University of Technology, Guangzhou 510640, China
2    Peng Cheng Laboratory, Shenzhen 518100, China
*    Correspondence: auhlpei@scut.edu.cn

**Abstract:** In this paper, we extended video stitching to nearshore bathymetry for videos that were captured for the same coastal field simultaneously by two unmanned aerial vehicles (UAVs). In practice, a video captured by a single UAV often shows a limited coastal zone with a lack of a wide field of view. To solve this problem, we proposed a framework in which video stitching and bathymetric mapping were performed in sequence. Specifically, our method listed the video acquisition strategy and took two overlapping videos captured by two UAVs as inputs. Then, we adopted a unified video stitching and stabilization optimization to compute the stitching and stabilization of one of the videos separately. In this way, we can obtain the best stitching result. At the same time, background feature points identification on the shore plays the role of short-time visual odometry. Through the obtained panoramic video in Shuang Yue Bay, China, we used the temporal cross-correlation analysis based on the linear dispersion relationship to estimate the water depth. We selected the region of interest (ROI) area from the panoramic video, performed an orthorectification transformation and extracted time-stack images from it. The wave celerity was then estimated from the correlation of the signal through filtering processes. Finally, the bathymetry results were compared with the cBathy. By applying this method to two UAVs, a wider FOV was created and the surveying area was expanded, which provided effective input data for the bathymetry algorithms.

**Keywords:** bathymetry; video stitching; UAV; background identification; cBathy

## 1. Introduction

Coastal zone mapping plays a crucial role in oceanography, and water depth is a critical parameter that can directly reflect nearshore topography. However, the nearshore topography will be frequently changed due to wave motion, extreme typhoon weather, and human activities. It incurs tremendous problems for coastal zone management, for which we need to measure the nearshore bathymetry on a timely basis [1]. An accurate and convenient bathymetry method is urgently needed to be put forward and applied to practical work.

A convenient method is to operate an unmanned boat equipped with a sonar system weighing several hundred kilograms and then, measure depth using the acoustic principle. The disadvantage of this method is that it requires a lot of human resources and financial resources each time and cannot meet the flexible operation and low-cost needs. In addition, the airborne light detection and ranging (LiDAR) system was first developed by the United States military and provides high-precision bathymetry in cleaning waters. It is also not suitable for the above needs because of expensive costs and unreasonable spatial resolution [2]. Furthermore, satellite imagery is used for coastline topographic mapping and depth estimation [3,4] but is challenging to operate. Pressure sensor array [5] also can estimate depth continuously by measuring swell propagation velocity, and it is expensive

to install these devices. Traditional methods for gathering bathymetric data include a small watercraft [6] outfitted with a real-time kinematic global positioning system (RTK-GPS) that measure depths accurately to a few millimeters in calm seas or a jetski [7] with an echo sounder attached behind it. In beaches with regular low wave times and easy access to the sea for small vessels, such a monitoring strategy can be carried out several times a month at most [8]. Consequently, finding more accessible alternatives to obtaining bathymetric data with the high spatial and temporal resolution is of significant importance.

More recently, some scholars paid attention to obtaining the information they wanted from coastal video imagery of waves based on the linear dispersion relationship [9–13], which is mathematically described by Equation (1).

$$\omega^2 = gk\tanh(kh) \tag{1}$$

where $\omega$ is the angular frequency ($2\pi/T$, $T$ means wave period, or $2\pi f$, $f$ means wave frequency), $g$ is the gravitational acceleration constant. $k$ represents wavenumber ($2\pi/L$, $L$ means wavelength). An additional parameter is wave celerity $c$ ($L/T$). Since the focus is on water depth information h, Equation (1) can be deformed to Equation (2).

$$h = \frac{c}{2\pi f}\text{arctanh}(\frac{2\pi f c}{g}) \tag{2}$$

Holman [14] creatively collected the wave propagation characteristics from coastal imagery and performed the nearshore depths. The water depth can be calculated as long as we know two of these ($f, c, k, L, T$).

The method based on coastal video imagery is regarded as a promising way to observe nearshore depth because of its low cost and easy operation. The critical point of this method is how to estimate the related parameters in Equation (1) or Equation (2) precisely. According to previous studies, there are two different ways to complete the process of variable estimation: the frequency method and the temporal method. The frequency method was first laid out by Holman [14] and was then refined in the cBathy algorithm [14]. Over the last few decades, many types of research [8,11,15] were inspired by cBathy in actual work. Additionally, video monitoring stations play an important role in bathymetric methods, and video-based algorithms were also extended [16–19]. Recent advancements in UAV technology and cost-cutting measures make it possible to use instruments designed for video monitoring stations at locations where measurements are needed but no video station is accessible, either due to a lack of a high vantage point or the necessity for only a single survey. The use of videos captured by UAVs to obtain bathymetry was previously investigated in recent publications [10,20–22].

Classical photogrammetry reconstructs 2-D topography from stacking aerial images and has existed as a field for quite some time. However, the method mentioned above is either based on a fixed camera for mapping or requires the UAV camera to hover at a certain height. The cBathy algorithm and the reference [22] require a certain observation time for sea level water movement, which can only provide the sea wave movement data with a fixed angle of view. That is to say, one flight can only carry out terrain inversion for a fixed area. If it is necessary to carry out bathymetric mapping for a long-distance coastline, the original method can only be used to carry out sectional sampling through separate fixed-point sampling, which greatly reduces the efficiency of mapping. Simultaneous mapping of multiple cameras is realized to obtain a wide field of view (FOV) in a single UAS [23] but it is not suitable for commercial UAVs with limited conditions, from which we were inspired and proposed another method.

The objective of this paper is to demonstrate that coastal bathymetry can be estimated from panorama video using video stitching and wave speed inversion algorithms applied to imagery from two UAVs transiting along the coast. The presented system enables the collection of mobile, short-dwell time series by background feature points identification on the shore. Additionally, the contribution of the paper can be concluded as follows.

1. The UAVs video stitching creates a wider FOV and improves the bathymetric mapping ranges;
2. The process of video stitching eliminates some of the rectification biases.

The rest of this paper is arranged as follows: in Section 2, the acquisition of video and video stitching is described. Then, pixel intensity signals are extracted from time stack imagery, and several signal processes are described in Section 3. Related parameters for bathymetry and algorithm performance are evaluated in Section 4. The discussion is provided in Section 5. Conclusions are summarized in Section 6.

## 2. Video Processing

### 2.1. Video Acquisition Scheme

According to the linear dispersion relationship, identifying relevant parameters in Equation (1) determines the final effect. After actual testing and data analysis, a suitable site needs to present a low-gradient slope in the intertidal area. The video acquisition work was selected in Shuang Yue Bay (22°35′43.1″ N, 114°52′35.3″ E), Huizhou, China, eventually, a sandy coast that faces the western Pacific Ocean. Figure 1 shows a satellite map of Shuang Yue Bay. This area has a beach coastline of about 5 km in the direction of the longshore and a remarkable horizon in the cross-shore direction to observe swells. The angle between the incoming swells' propagation direction and the coastline is almost 0. The waves on this site had perfect motion characteristics, such as clear crest lines and ideal amplitude. Thus, we chose the bay as this work's image acquisition site and obtained several UAV videos of shallow water depth areas in this place under varying conditions from January to May 2022.

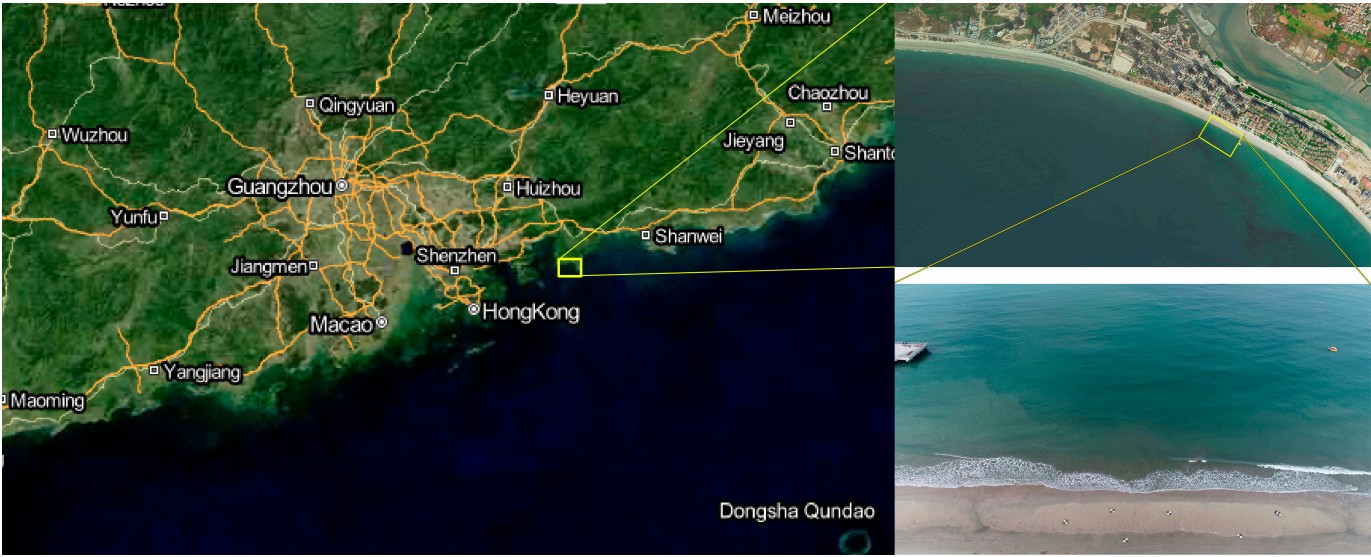

**Figure 1.** Shuang Yue Bay satellite map and image.

We used the UAV of DJI Phantom 4 RTK, equipped with a normal wide-angle lens with 24 mm focal length, which was based on a 20 M pixel CMOS chip. The video resolution of the UAV camera was 4 K *UHD TV* (3840 × 2160). However, the feature of the camera is that it supersampled the frame in record state rather than photo state, reducing the FOV from 84° to 80.13°. To simulate small commercial UAVs, we reduced the pixel resolution to *FHD* (1920 × 1080). Due to a continuous video being required, the frame rate of the captured video was set to 30 fps. The RTK refresh rate was too low to represent the exact position of the camera. Commonly, relying on the traditional theoretical method [24], we placed several ground control points (GCPs, Figure 2) on the beach to solve the extrinsic parameter of the camera.

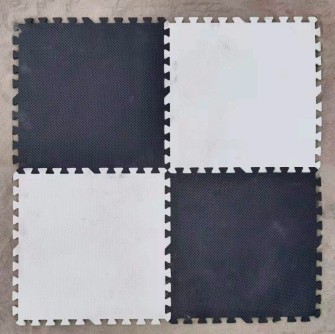

**Figure 2.** Ground control point.

The formula between camera parameters and the coordinate system can be shown in Equation (3),

$$p_{uv} = KTp_w \tag{3}$$

where $K$ is the camera intrinsic matrix that contains focal lengths and principal point, $T$ the camera extrinsic matrix containing rotation matrix $R$ and translation vector $t$, depending on camera orientation and location, often defined as $T = \begin{bmatrix} R & t \\ 0^T & 1 \end{bmatrix}$. $p_{uv}$ and $p_w$ represent the coordinates of the pixel coordinate system and world coordinate system, respectively. The yaw angle of the camera should face the wave propagation to record swell motion correctly. In addition, the intersection angle between this camera's yaw angle and the wave propagation direction should not exceed 75° because an optical camera cannot capture the crest line [25]. Therefore, the yaw angle was set to approximately 30~45° relative to the region of interest to meet the above requirements. As a rule of thumb for depth inversion video collected processes, UAV should be deployed as high as possible to ensure the algorithm's field of view and better effect. Bearing this in mind and ensuring the UAV did not exceed a safe and controllable altitude, we decided to operate the UAV hovering at the height of 60~120 m. The video duration ranged from 4 to 15 min.

### 2.2. Image Processing

The core data required by the algorithm were to obtain the time series of the location where the water depth must be measured. Thus, these captured videos should be converted into sequential orthoimages that can conveniently extract time series precisely. It is unnecessary to preserve every frame for a high frame-rate video because of increasing computation. We are setting an appropriate frequency for down-sampling operation and retaining information of keyframes only. In this paper, the sampling frequency was set to 2 Hz, which meant that one image frame was extracted in 0.5 s from videos and combined into a complete-time sequence. The down-sampling operation would not affect the extraction of crucial information about waves because the periods of offshore waves were about 10 s.

As per the hardware structure design of the camera, there would be more or less distortion in the video and photo during recording. Camera calibration aims to obtain the intrinsic parameters and distortion coefficient for subsequent image transformation. This process refers to the method mentioned in [26].

### 2.3. Video Stitching and Stabilization

With a fixed angle of view, the video captured by an UAV for bathymetry was often limited by the camera's field of view (FOV). Stitching the video is a way to increase the horizontal FOV of the camera, and adding another camera appropriately reduces the influence of the rectification bias of the first camera. In this paper, bundled video path and path optimization will be used for video stitching. The details of this algorithm will be described below.

To stabilize an input video, Liu et al. [27] divided the video into grids spatially, as shown in Figure 3. A single homography $F_i(t)$ is estimated between neighboring frames in the original video, where $i$ represents each grid at frame $t$. It is estimated by tracked features between adjacent frames. The camera path can be defined as the multiplication of a series of continuous homography.

$$C_i(t) = F_i(t) \cdot F_i(t-1) \cdots F_i(1), 1 \leq t \leq T, 1 \leq i \leq m^2 \tag{4}$$

where $T$ is the number of frames in the video file and $m^2$ is the size of the m-by m grids. Local changes in the gird usually have a better transition to the stitching effect. The camera path can first be computed the feature trajectories by KLT tracker [28]. Given the original path $C = \{C(t)\}$, what needs to be calculated is the optimized video path and it is defined as the optimized path $P = \{P(t)\}$:

$$\Theta(P_i) = \sum_t \left( ||P_i(t) - C_i(t)||^2 + \lambda \sum_{r \in \Omega_t} \omega_{t,r} \cdot ||P_i(t) - P_r(t)||^2 \right) \tag{5}$$

where $\Omega_t$ denotes the neighborhood at frame $t$. The data term $||P_i(t) - C_i(t)||^2$ enforces the optimized path to be close to the original one, and the next term $||P_i(t) - P_r(t)||^2$ mainly stabilizes the optimized path. $\lambda$ and $\omega_{t,r}$ balances the two terms. $\lambda$ empirically set to 5. $\omega_{t,r}$ can be calculated by two Gaussian functions [27]:

$$\omega_{t,r} = G(||r - t||) \cdot G(||C_i(r) - C_i(t)||) \tag{6}$$

If all the grids participate in path optimization, the above Equation becomes:

$$E^{stable}(P) = \sum_t \left( \Theta(P_i) + \sum_t \sum_{j \in N(i)} ||P_i(t) - P_j(t)||^2 \right) \tag{7}$$

where $P = \{P_i | 1 \leq i \leq m^2\}$, $j \in N(i)$ means the grid $j$ is the neighbor of the gird cell $i$. According to the reference [29,30], $E^{stable}$ works like a stabilizer that reduces shakiness during stitching. Additionally, $P$ can be obtained by iteration [27,31], whose initial value can be set as $P = C$. Recall that video stitching is to create a wider FOV. $P^A$ and $P^B$ are the optimized paths generated by video A and video B, respectively, $H$ is a single homography used to stitch the two videos. As for the process of stitching, there is the following optimization function that achieves stitching and stabilization at the same time:

$$E(P^A, P^B, H) = E^{stable}(P^A) + E^{stable}(P^B) + \beta \cdot E^{stitch}(P^A, P^B, H) \tag{8}$$

$$E^{stitch}(P^A, P^B, H) = \sum_t \sum_k \left|\left| P_{\bar{i}}^A(t) \cdot C_{\bar{i}}^A(t)^{-1} \cdot v_k^A(t) - H \cdot P_{\bar{j}}^B(t) \cdot C_{\bar{j}}^B(t)^{-1} \cdot v_k^B(t) \right|\right|^2 \tag{9}$$

where $v_k^A(t)$ and $v_k^B(t)$ are the $k^{th}$ feature point calculated by SIFT [32] at frame $t$ of video A and video B, respectively. $\bar{i}$ and $\bar{j}$ are the grids where the feature points $v_k^A(t)$ and $v_k^B(t)$ are located. $E^{stitch}(P^A, P^B, H)$ means using SIFT to stitch frames. $\beta$ was set between 10 to 30, depending on the parallax between the two videos.

In our study, the videos of two UAVs were spliced into a panoramic video. However, the above video path optimization process will more or less change the camera's optical center position and distortion degree, similar to as-projective-as-possible image warping in the method [33]. The frames should be transformed as little as possible to maintain pixel information integrity. The improvement of this paper is that the path of one of the videos is

not optimized, which can also reduce the amount of calculation. Assume that the path of video A is not optimized, Equation (8) becomes:

$$E(P^A, P^B, H) = E^{stable}(P^B) + \beta \cdot E^{stitch}(P^A, P^B, H) \tag{10}$$

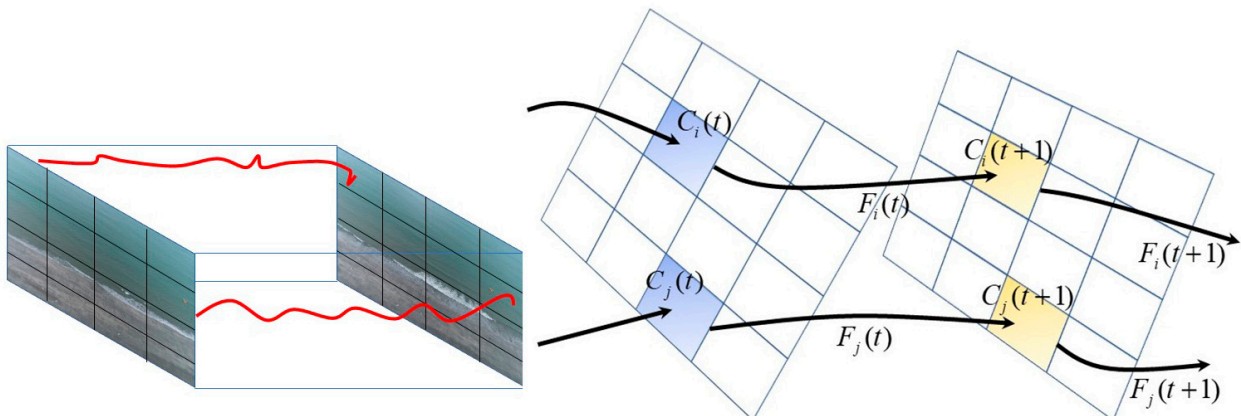

**Figure 3.** Each grid has its own bundled path, consisting of a sequence of homography.

The original Equation (7) optimizes the path of two videos. In this process, the optimized paths will be stabilized and remove shakiness first, then the feature matching method will be used for stitching. Note that $P^A$ and $P^B$ are also influenced by the stitching term. By setting $P^A(t) \equiv C^A(t)$, $P^B(t)$ tends to $P^A(t)$, i.e., the right frame B absorbs all perspective distortions (Figure 4). With known $P^A(t)$ and $H$, we can only calculate $P^B(t)$ by iteration. In theory, with the number of iterations $\xi$ increasing, the camera $B$ extrinsic matrix (in Equation (3)) is tending to the camera $A$ extrinsic matrix:

$$\begin{cases} \lim\limits_{\xi \to \infty} R_B^{(\xi)} = R_A \\ \lim\limits_{\xi \to \infty} t_B^{(\xi)} = t_A \end{cases} \tag{11}$$

Based on the above principles, we planned two flight routes, and the distance between two UAVs was reasonably adjusted from 100 m to 130 m according to the lens parameters of the cameras. All the waypoint action setting was completed on the software DJI terra. As for other parameters such as $\lambda$ and $\omega_{t,r}$ in Equation (5), $\beta$ in Equations (8) and (10), we set and performed them the same as reference [30]. Table 1 shows the flight parameters of the two UAVs. Figure 5 demonstrates the flight routes and panoramic results of stitching.

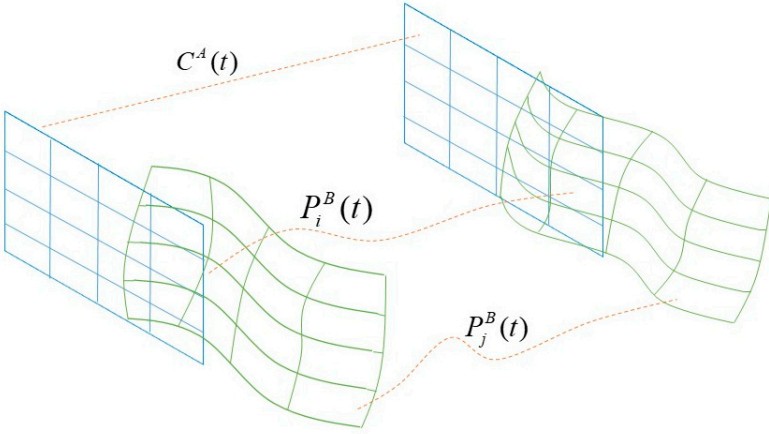

**Figure 4.** The process of unified stitching and stabilization.

**Table 1.** The flight parameters of two UAVs.

| UAV Number | Height (m) | Camera Yaw (°) | Camera Pitch (°) | Linear Speed (m/s) |
|:---:|:---:|:---:|:---:|:---:|
| A | 79 | −148.5 | −35 | 0.2 |
| B | 80 | −148.5 | −35 | 0.2 |

*2.4. Orthorectification and Background Identification*

An essential step in this algorithm is orthorectification to sequence the images sampled in the previous section.

The orthorectification transforms imagery into maps that remove remote sensing equipment and terrain-related geometric distortions [34]. We extracted the region of interest (ROI) frame by frame to achieve this goal. Thus, the orthorectification of each image consisted of mainly four stages.

(1) The first step is to determine the real-world coordinates of ROI by RTK-GPS;
(2) The second step is to determine the pixel resolution;
(3) The third step is to calculate the ROI pixel coordinates using GCPs;
(4) The last step is to reorganize these pixels into a complete image for the algorithm's input.

It is worth mentioning that the extrinsic parameters of each image need to be solved once because the UAV may be affected by its mechanical vibration and environmental factors such as wind-related shaking. There were six parameters required to be estimated in Equation (3), that is Equation (11), position information $(x_w, y_w, z_w)$, and camera Euler angles (roll, pitch and yaw, shown in Figure 6) in $R$.

$$z_c \begin{bmatrix} u \\ v \\ 1 \end{bmatrix} = KR \begin{bmatrix} I & t \end{bmatrix} \begin{bmatrix} x_w \\ y_w \\ z_w \\ 1 \end{bmatrix} \tag{12}$$

We use black-and-white sheets as GCPs to identify them in each frame quickly. For some UAVs with simple equipment and a low price, the data of the inertial measurement unit (IMU) and GPS positioning module carried by them may not be the most accurate, which often leads to difficulties in estimating camera extrinsic parameters, especially on frames with an insufficient number of GCPs due to disappearance of GCPs during UAV movement. In the subsequent recognition, with static or simple camera motion, the pixel coordinates of GCPs are determined by color threshold classification or template matching. Considering the complex camera motion, background identification [35] is probably an effective method (Figure 7). The motion of the background points can be closely approximated to the camera motion by clustering and recognizing them on the feature stream of continuous frames. As a visual odometry, the product of the homography matrix $H$ computed from the background points between frames is temporarily used to estimate the camera motion path $M$, which is similar to Equation (4).

$$\begin{cases} M(t) = H(t) \cdot H(t-1) \cdots H(1), 1 \le t \le T \\ M(t) = H(t) \cdot M(t-1) \end{cases} \tag{13}$$

where $T$ represents the number of frames in a video file. Estimate the camera pose by the above formula, then list the values of the world coordinates and calculate the pixel coordinates to resample the frames by using Equation (12).

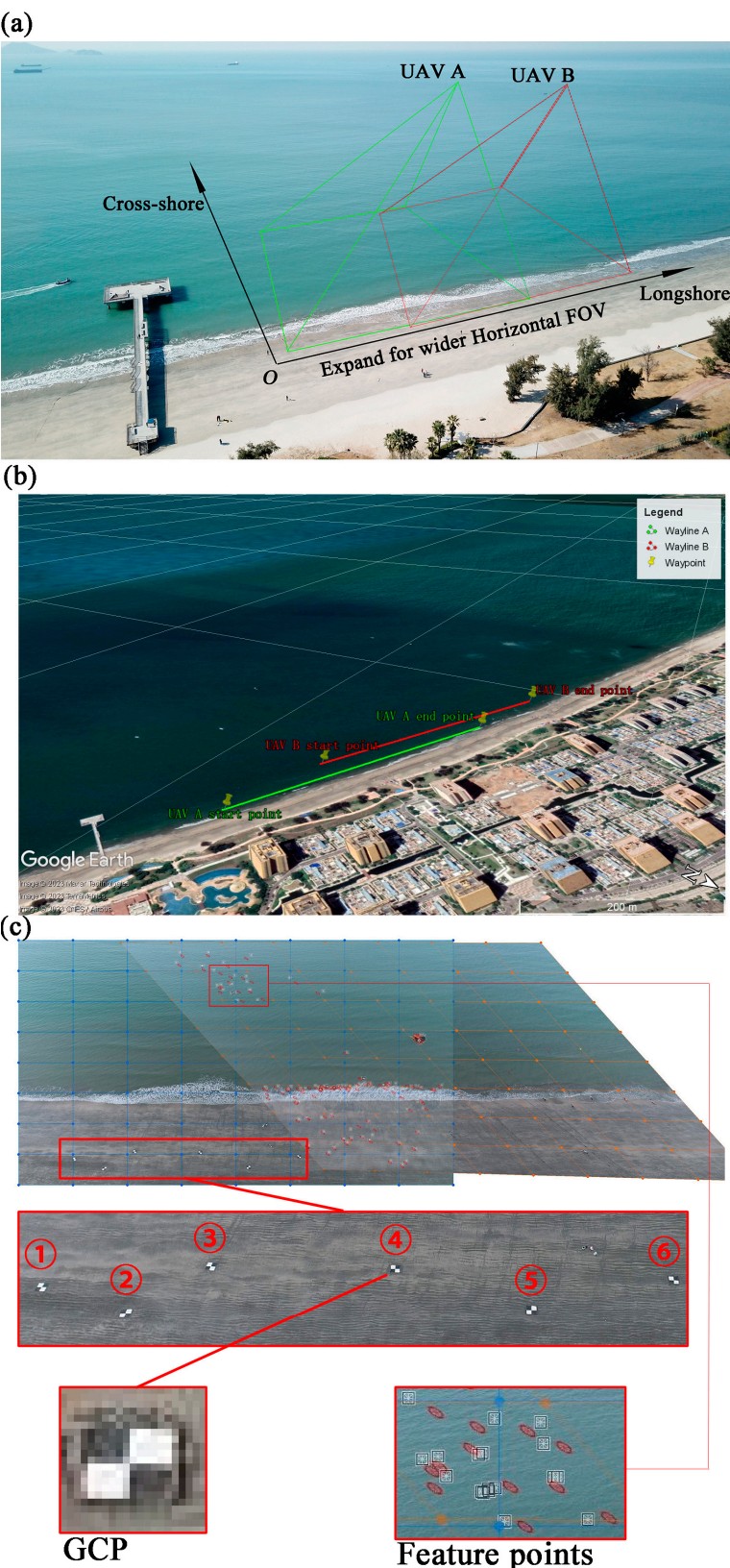

**Figure 5.** Schematic diagram of flight routes. (**a**) Planning two UAVs to create a wider horizontal FOV. (**b**) The flight routes of the two UAVs. (**c**) The panoramic results of stitching with grids. We placed six GCPs on the beach. The white squares and the red circles are the feature points of the left and right videos, respectively.

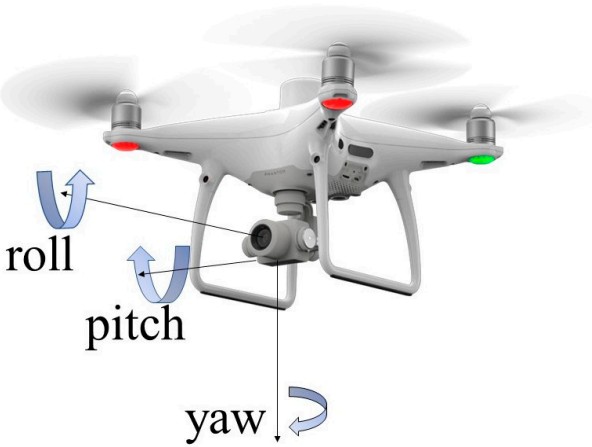

**Figure 6.** Camera Euler angles.

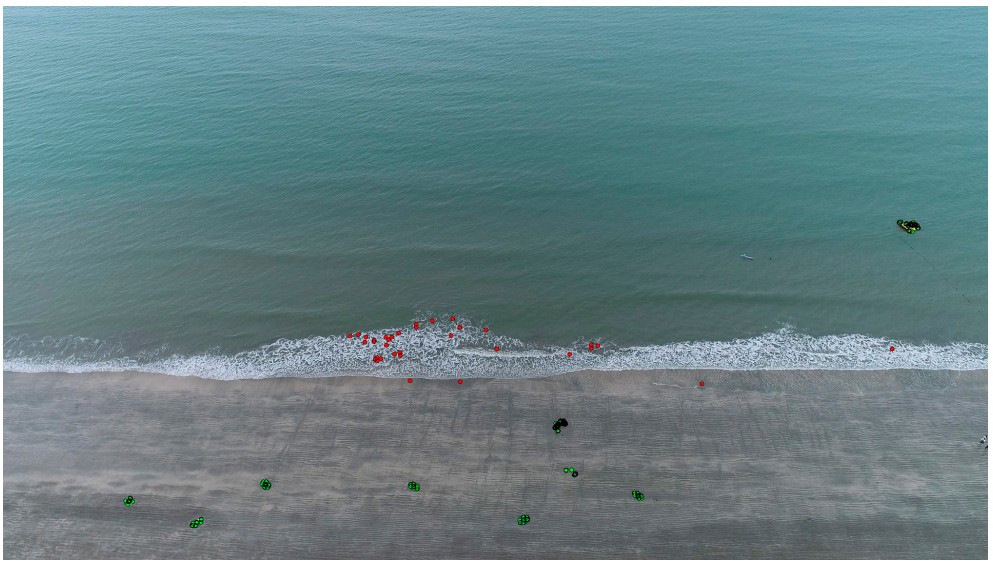

**Figure 7.** Background identification. The green points are marked as background points, while the red points are marked as foreground points.

All steps are shown in Figure 8 for a single UAV. We chose a GCP as the origin of the world coordinate system and determined the pixel area corresponding to the whole ROI. Table 2 demonstrates the configuration of orthoimages.

(a)                                                                                              (b)

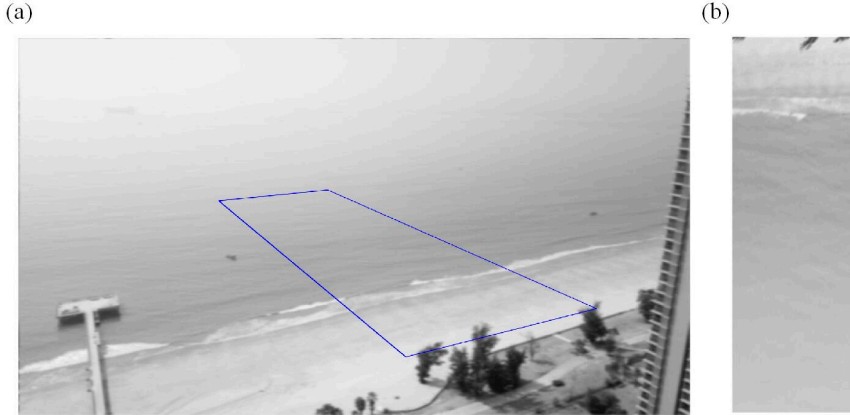

**Figure 8.** Schematic diagram of interest region selection. (**a**) The target area is shown in the blue line in the original scene image. (**b**) Unified pixel resolution orthoimage.

**Table 2.** Configuration of orthoimages of a single video.

| Pixel Resolution | Cross-Shore Range | Longshore Range |
|:---:|:---:|:---:|
| 0.5 m | 0~200 m | 0~100 m |

## 3. Signal Extraction

### 3.1. Time Stack-Based Pixel Intensity Signal

In the previous section, we solved the orthorectification of down-sampling images to ensure strong consistency in pixel resolution. The motion characteristics of waves can be directly reflected by the pixel fluctuation. Hence, the pixel intensity is the unique identification of the wave signals in this work.

The pixel intensity signals can be extracted from a series of orthorectification images. Figure 9 shows the time stack image of one specified transect.

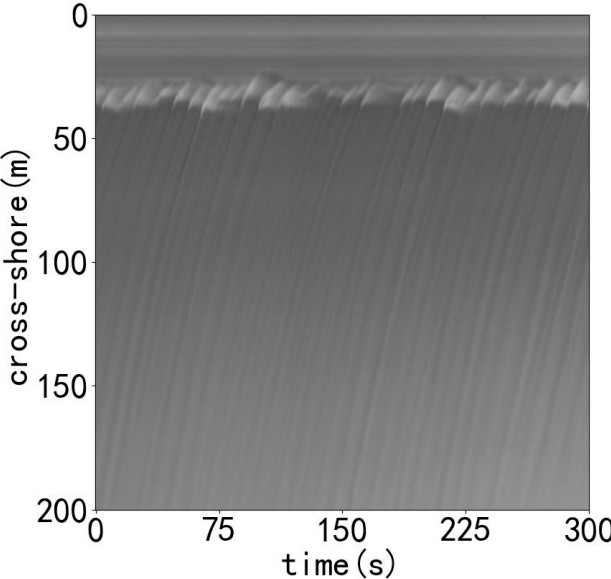

**Figure 9.** Time stack image in one specified transect.

### 3.2. Filtering Process

Ideally, a swell can be approximated as a superposition of multiple standard sine waves. However, many short wind waves unrelated to swell movement exist on the sea surface. It means that the pixel intensity signals are mixed with noise independent of water depth. Additionally, it is difficult to directly obtain the available signal from the down-sampling images under the joint action of various other uncertain factors without properly handling by analysis of raw signals. Therefore, we filtered the signal three times to obtain the signal component linked to the water depth.

The first filtering stage eliminates high-frequency noise such as short wind waves by an image filtering method. Figure 10 shows that the Gaussian low-pass filter significantly attenuated pixel intensity fluctuations. The first step of filtering separates the pixel-intensity signal from high-frequency noise. Undoubtedly, the noise filtering process enormously increased accuracy in this study.

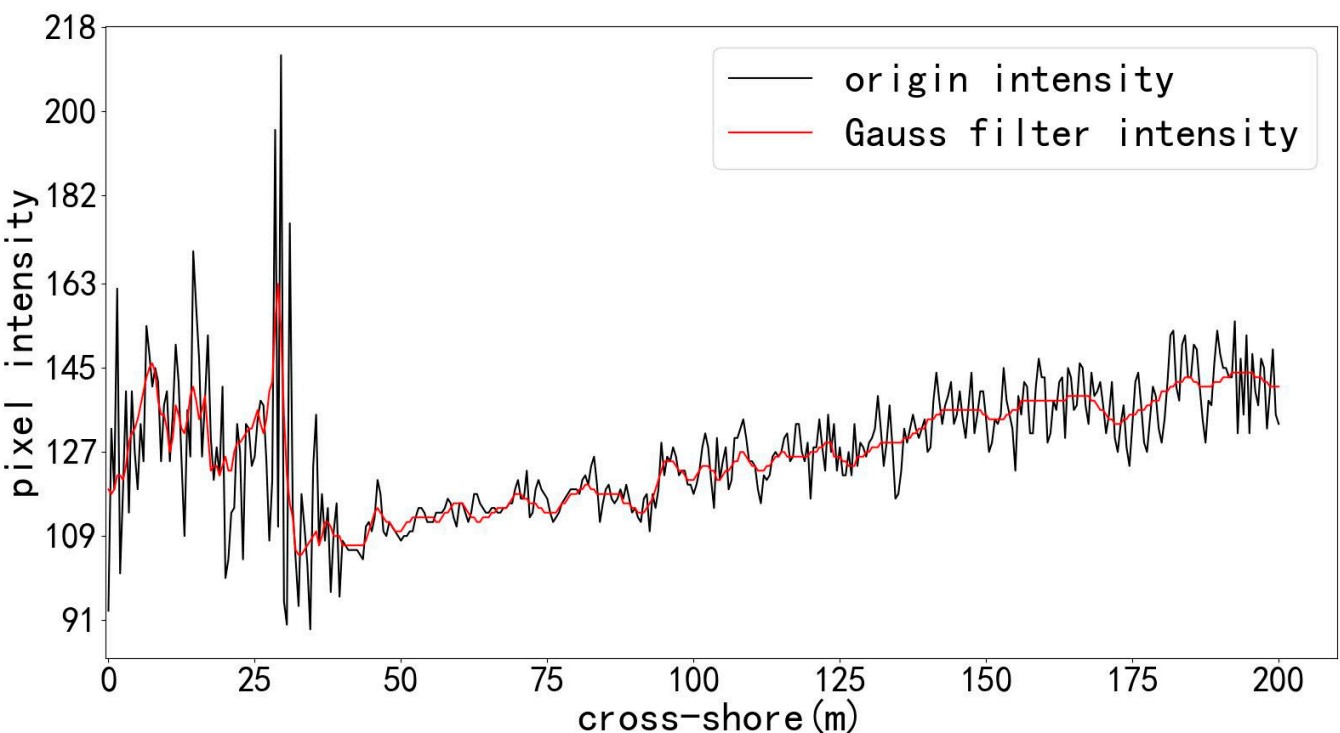

**Figure 10.** Comparison between Gaussian low-pass filtered signal and the original signal.

The second filtering stage desires to get rid of the influence of other irrelevant factors. A band-pass filter will cut the pixel intensity signal with a frequency range of 0.05~0.5 Hz. As shown in Figure 11, we analyzed the pixel intensity signal based on time-stack images by the fast Fourier transform. We found that the main components were concentrated in the above frequency bands. It can be considered that the frequencies outside this range were independent of swell waves [36]. These irrelevant frequency components may be generated by breaking waves and optical noise.

The last but crucial step was to purify the frequency components of time-stack signals. After a 0.05~0.5 Hz band-pass filter process, the signal will still contain a great variety of useless features. Selecting several representative frequency components can reduce the amount of computation and highlight the correlation between different signals. The principle of choosing the dominant frequency is based on the cross-correlation in a wavelength range. A correlation analysis method inspired by [14] was adopted in this work. We enumerated some possible frequency bands in advance according to frequency distribution. As stated by Equation (14), a cross-spectral matrix was computed between all pixel intensity signals within a wavelength range in a cross-shore direction for each frequency band.

$$C_{ij}(f) = \left\langle \hat{G}(x_i, y_i, f)^*, \hat{G}(x_i, y_i, f) \right\rangle \tag{14}$$

where superscript * indicates the complex conjugate. The Fourier transform of the intensity signal at one pixel would be performed in Equation (15).

$$G(x, y, f) = FT(S_{pixel}(x, y, t)) \tag{15}$$

Subsequently, we created the coherence squared for all the potential frequency bands based on the cross-spectral matrix to determine which band had the most significant impact on signals. The bands that contributed the most to the cross-spectrum will finally be retained.

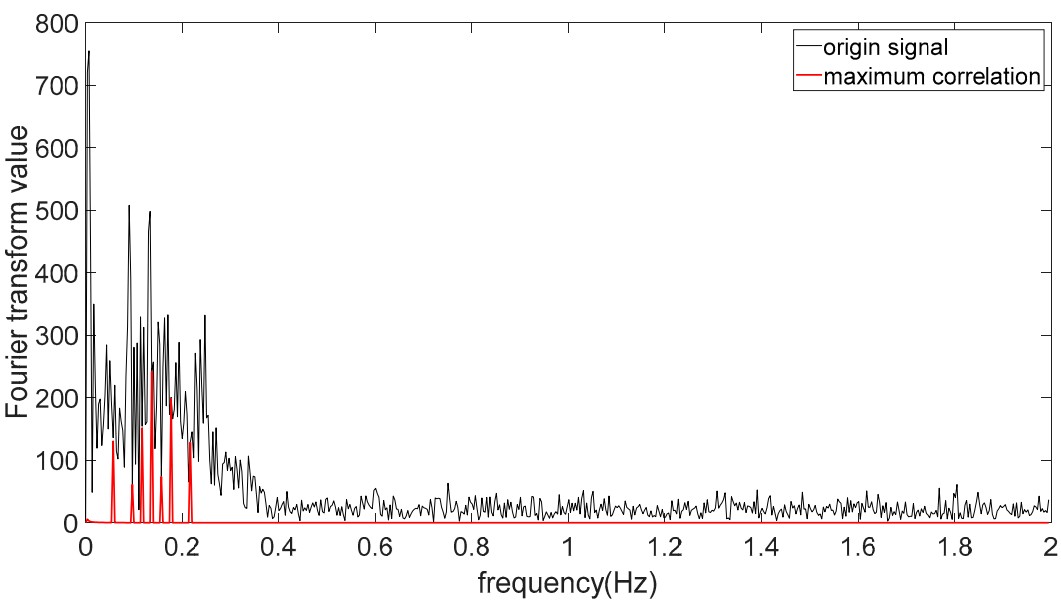

**Figure 11.** Frequency distribution of raw pixel intensity signal.

## 4. Bathymetry Results

In this paper, our bathymetric mapping method mainly referred to the estimation theory of wave celerity [5], the contributions made by Lippman and Holman in determining the dominant frequency of ocean waves, and the method of Tsukada [22]. Moreover, the method in this paper is called timeCor since this method's main signal processing flow is in the temporal domain and uses the cross-correlation analysis method between signals. Meanwhile, the bathymetry mapping results of temporal cross-correlation analysis (timeCor) will be compared with those of the cBathy algorithm [14].

### 4.1. Wave Celerity and Frequency Estimation

In this study, the direction of wave propagation was considered perpendicular to the coastline after orthorectification. The wave celerity was estimated within a suitable range in the cross-shore direction. Firstly, we determined the optimal range of wave celerity estimation according to whether it can reflect a complete propagation characteristic of swells. We used a time delay method to determine the range. Concretely, an empirical time lag $\Delta t$ was fixed to 3 s. Then, we chose a reference pixel location $i$ such that the pixel at 150 m offshore in Figure 12a (the red line), all neighboring pixel locations $j$ from 1 to $i-1$ (0~149 m) participated in the calculation of the correlation coefficient according to Equation (16).

$$Cor(x_{ij}, y_{ij}) = \langle I(x_i, y_i, t), I(x_j, y_j, t + \Delta t) \rangle \tag{16}$$

As shown in Figure 12b, for each pixel in the cross-shore direction, the range between the closest point with the maximum positive correlation coefficient (the white line) and reference pixel is regarded as the suitable range for wave celerity estimation. The wave celerity of the reference point was evaluated from 134 to 150 m in Figure 12a. We calculated the cross-correlation coefficient between all pixels and the reference point in this range to determine the propagation time. Then, wave celerity was confirmed by the linear fitting method. The slope of the fitting line is regarded as wave celerity. Figure 13 illustrates the wave celerity estimation process.

According to Equation (2), the represented frequency of swell should be known before obtaining the water depth. The weighted average method Equation (17) was used to calculate the representative frequency of each wave. The variable $i$ was configured from 1 to $N$, which means that the power spectrum value of each valid frequency band would be weighted once.

$$f_{rep} = \frac{\sum\limits_{i=1}^{N} S_i f_i}{\sum\limits_{i=1}^{N} S_i} \qquad (17)$$

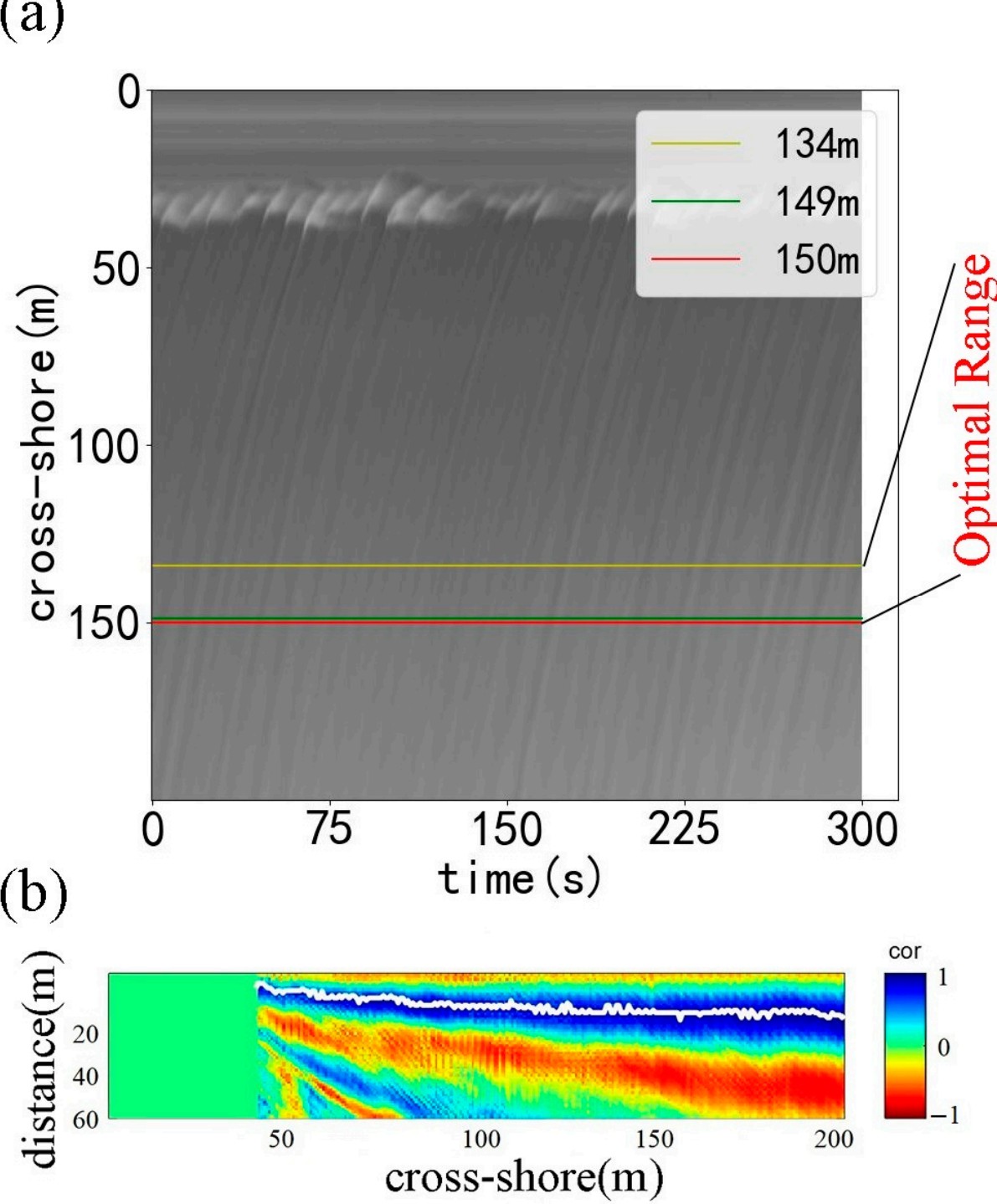

**Figure 12.** Optimal range estimation. (**a**) The cross-correlation calculation is carried out with this reference point (the red line) towards the shore. An optimal range is determined. (**b**) The relationship between pixel intensity cross-correlation and distance in the whole region.

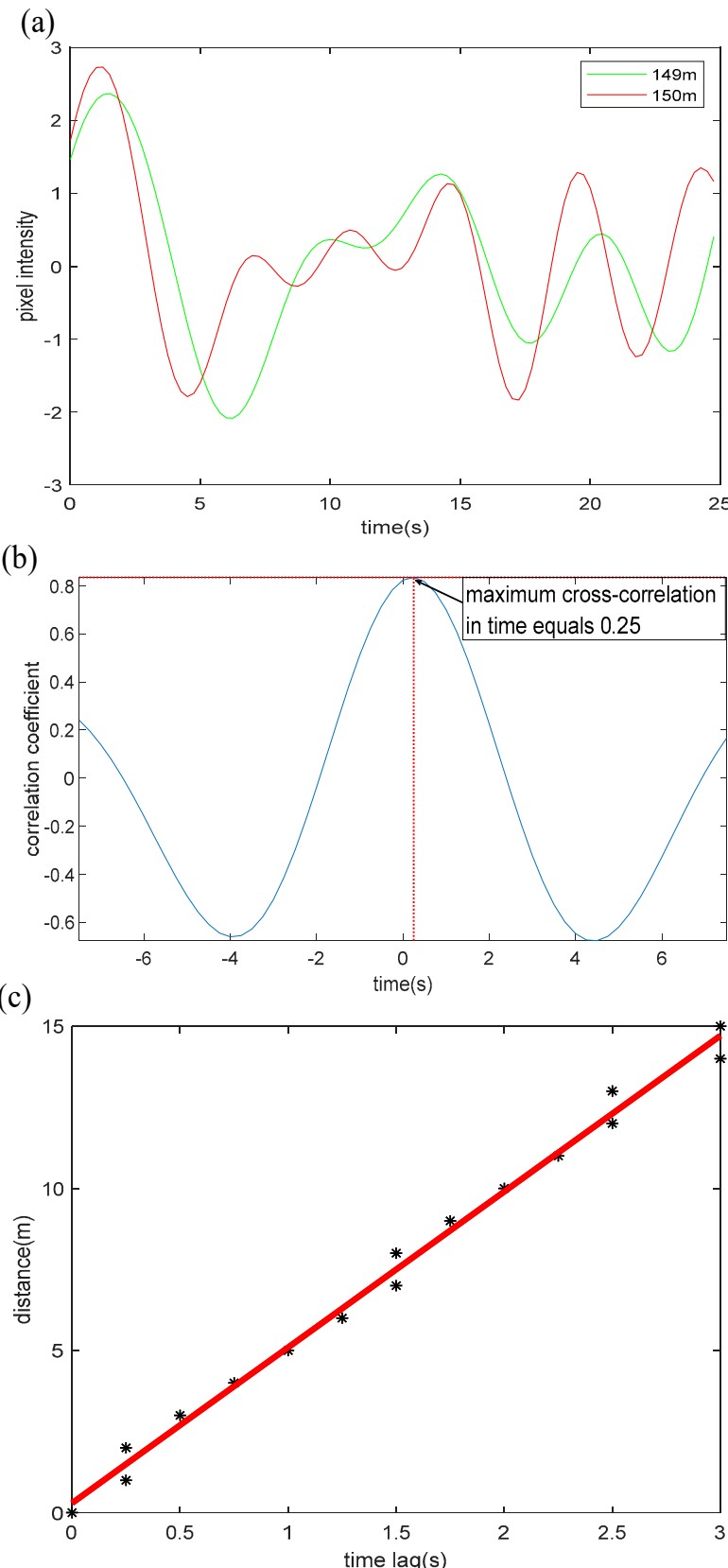

**Figure 13.** Wave celerity estimation. (**a**) The pixel fluctuation of adjacent pixels shows a specific correlation. (**b**) Maximum cross-correlation determines the propagation time of the wave. (**c**) The propagation time and distance between each pixel and the reference pixel in the suitable estimation range are determined and carried out the linear fitting method.

### 4.2. Bathymetry Result for A Single UAV

We used the unmanned remote-control boat with RTK-GPS and single beam sonar system to carry out the task of bathymetry to regard as the ground true. We chose two transects to evaluate the final effect. Figure 14 demonstrates the result for a single UAV based on Figure 8 and Table 2.

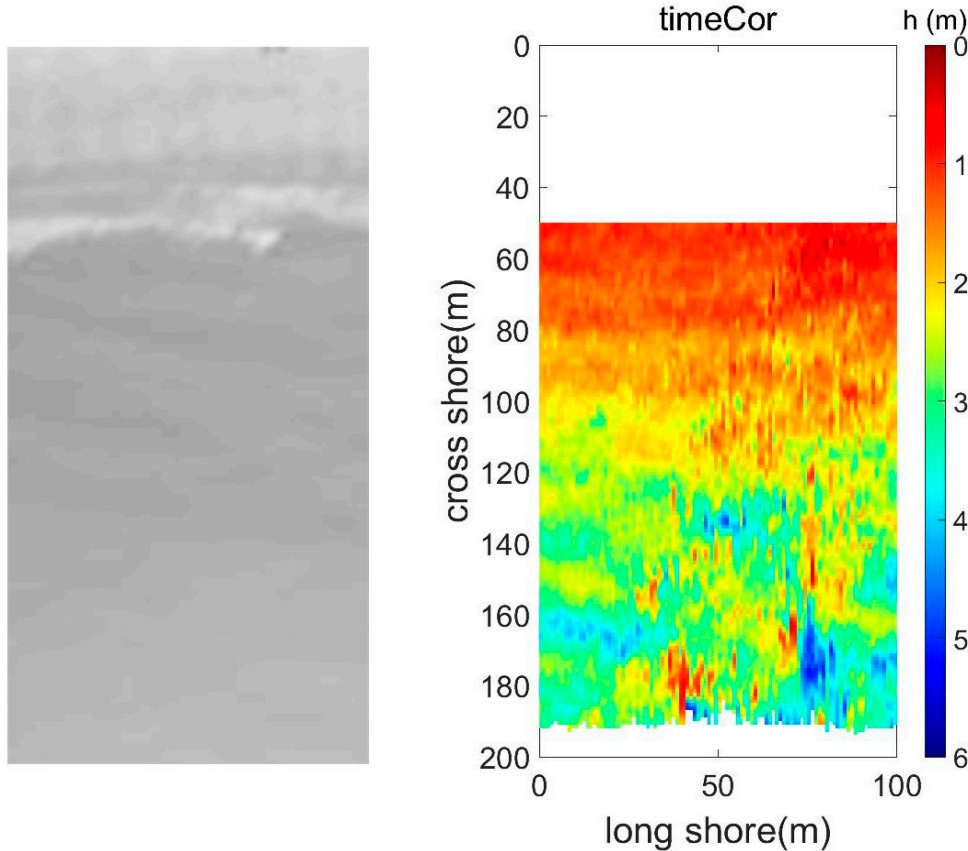

**Figure 14.** Bathymetry results of the target area, which longshore is 100 m and cross-shore is 200 m.

We evaluate some reference transects longshore with ground truth by boat in Figure 15. It can be seen that the error mainly comes from far offshore. The reason is that this method estimates in a one-dimensional direction, and characteristics of pixel intensity are obviously not captured at sea surface far away from shore.

In this paper, we compared the results of the cBathy algorithm and the method in Figure 16. The depth estimation area of the cBathy algorithm cannot be set prior to analysis, which will cause a significant error when the correlation of pixel signals is analyzed. An unreasonable selection of frequency bands will also affect the bathymetry. Additionally, the source of error mainly comes from the difference between each algorithm. The principle of timeCor is based on the degree of cross-correlation between signals, further analyzing the cross-power spectrum, calculating the energy distribution and correlation coefficient, and estimating the wave celerity and main frequency. The difference from cBathy is that each estimation is only performed in the one-dimensional cross-shore direction, as a result of which its result is less robust than cBathy one. Additionally, cBathy estimates in two dimensions. However, timeCor's calculation process is simple, and the whole process does not need to set too many parameters, which is suitable for temporary single estimation along the coast. A single cBathy depth estimation is often affected by external factors such as environment and acquisition, which leads to the inability to reverse the appropriate depth. The algorithm provides a Kalman filter link. If there is long-term observation data, the area that cannot be estimated at a single time can be compensated by the observation value and prediction value, but this paper is only based on a single estimation. Actually,

in the waters far away from the coastal zone, the UAV camera did not obtain relatively complete pixel intensity features, or at a certain angle, the pixel feature will be more or less affected by the camera's angle of view. Hence, the comparison between the two is only to confirm the consistency of the bathymetry results rather than determine which is better.

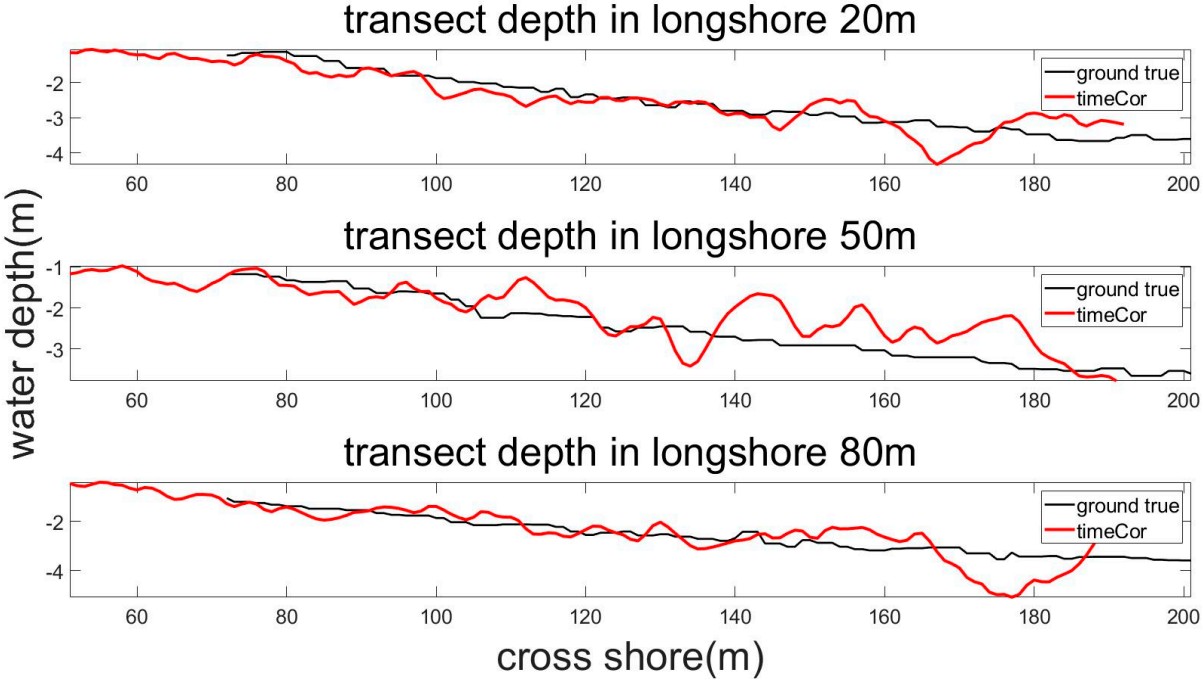

**Figure 15.** Comparison between ground true and the timeCor result.

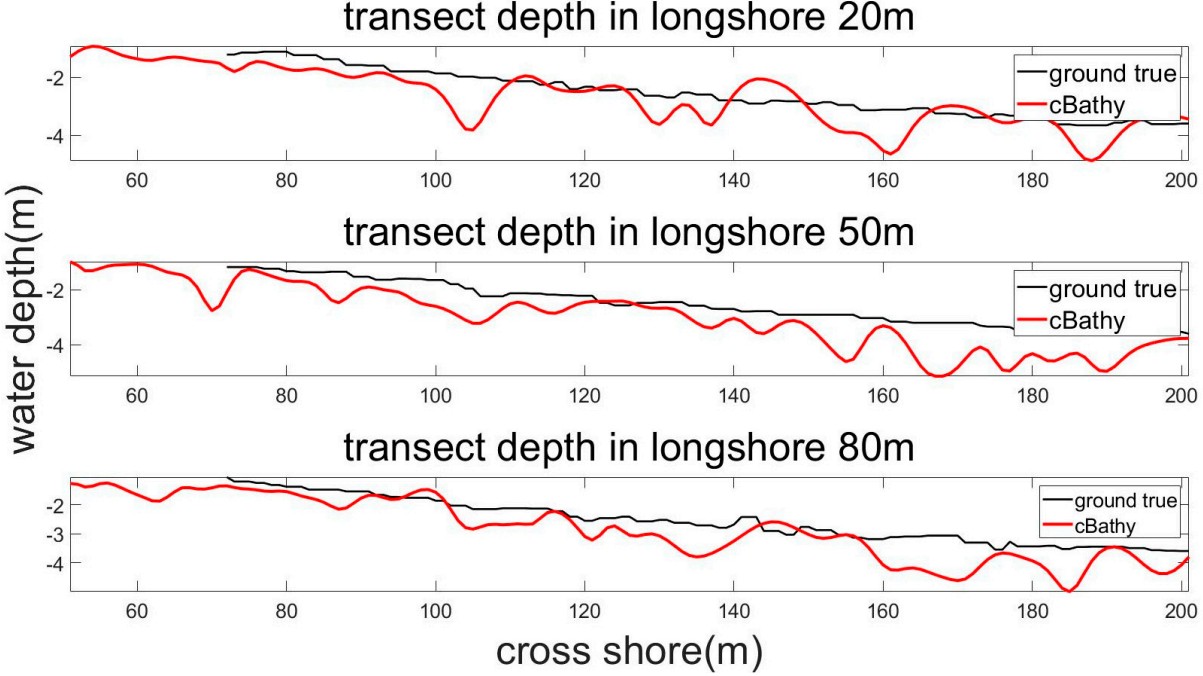

**Figure 16.** Comparison between ground true and the cBathy.

*4.3. Stitching Result*

From the previous section, we can see that the mapping view of a single UAV was limited. To achieve dual UAV bathymetry, we adjusted the placements of two UAVs. The validation of the above theory in Section 2.3 was carried out for the selected flight

parameters. With the same bathymetric algorithms, we expanded the ROI area (Figure 17) and obtained considerable results, whose longshore ranged from 0 m to 200 m. In fact, our boat did not measure the real water depths over such a wider range. The purpose of this paper was more to demonstrate the effectiveness of video stitching rather than the accuracy of bathymetry algorithm (timeCor). Therefore, we considered the results of the more robust cBathy to be closer to the ground truth in this subsection. Figure 18 displays the bathymetric mapping water depths of the two algorithms. The convergence of the two algorithms was in good agreement with the experimental results due to the fact that they were both based on the water depth solved by the linear dispersion relationship, although there are some differences in the core and calculation. Moreover, we compared the depth curves of the two (Figure 19), and the mean absolute error (*MAE*) was used to describe the similarity of the depth curves. The invalid points in the curves were eliminated. The smaller the *MAE* was, the more similar the two curves are. Meanwhile, the similarity function *MAE*(*longshore*) was drawn to represent the overall difference between the two results. Additionally, *MAE* is defined as:

$$MAE = \frac{1}{n}\sum|h_1(x) - h_2(x)| \tag{18}$$

where *n* is the number of the sum of the valid points, *h* means the water depth, and *x* represents cross-shore distance.

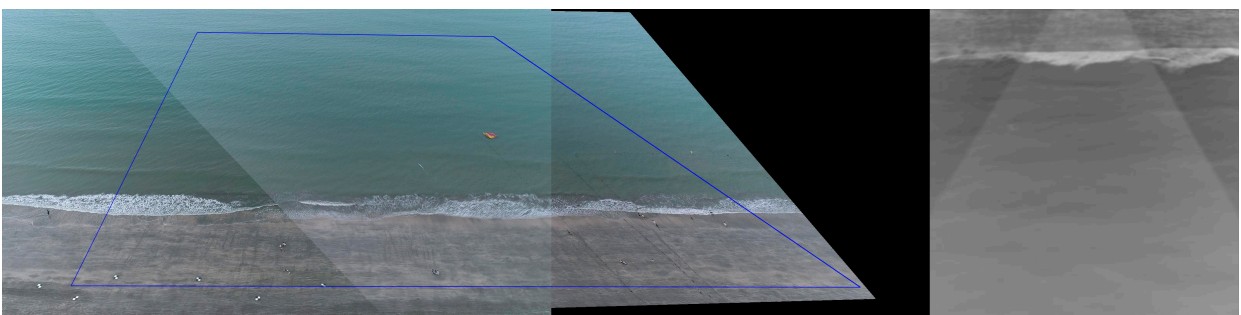

**Figure 17.** The range of ROI (200 m × 200 m) and its orthoimage.

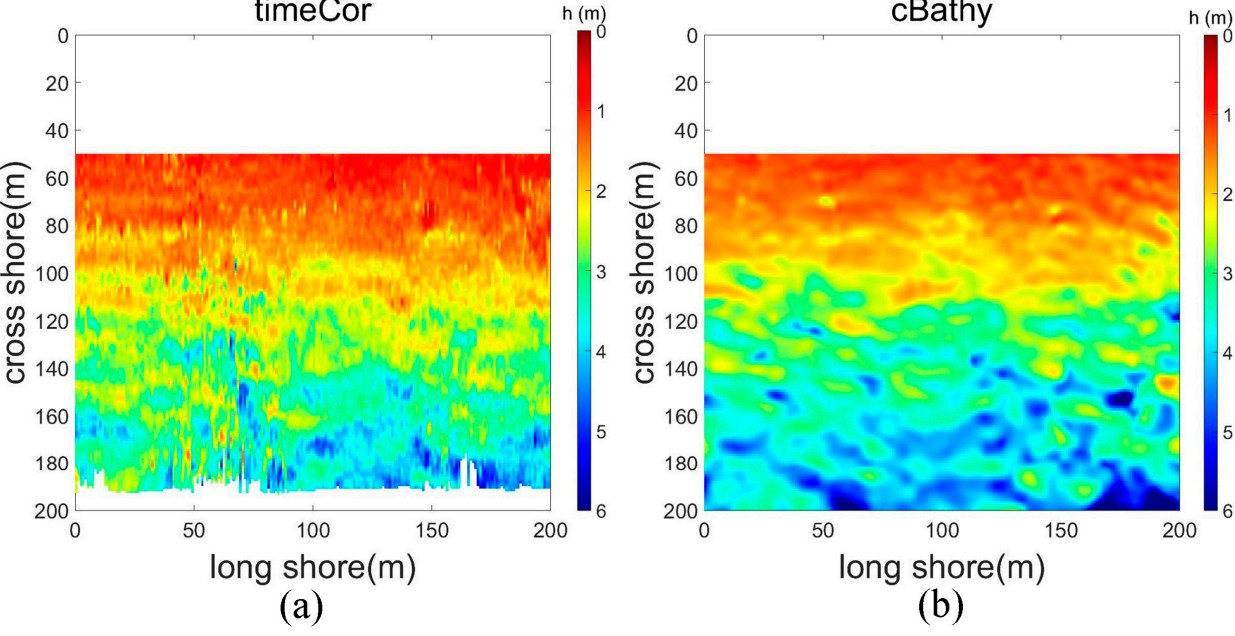

**Figure 18.** Bathymetry results of timeCor (**a**) and cBathy (**b**).

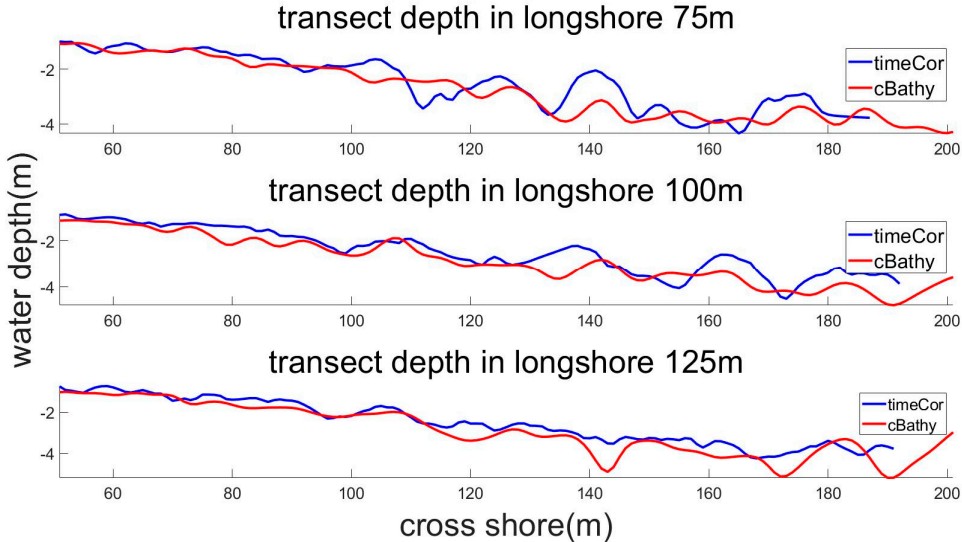

**Figure 19.** Comparison between timeCor and cBathy in longshore 75 m, 100 m, 125 m.

We calculated the global mean square error (MSE) and the global root mean square error (RMSE) with the two results, whose values were 0.37 m$^2$ and 0.60 m, respectively. It can be seen that the depth curves at about 50 m and 200 m longshore were very inconsistent in Figure 20. A persuasive conclusion supported by [22] is that there exhibits an underestimation bias in marine areas, which results from the remote distance from the camera (Figure 21). As a result of the addition of a camera during the process of video stitching, the rectification bias of the other camera would, to some extent, be compensated. For instance, the far-side swell waves often affect the results error of camera A alone, whereas camera B typically relies on its distance from camera A to compensate for the rectification bias of camera A. Thus, to further limit the influence of rectification bias while capturing videos, camera Euler angles and camera distances should be adjusted according to the field site. Meanwhile, the image blending strategy can be appropriately abandoned if the videos are severely out of synchronization. Generally speaking, our video stitching method can create a wider FOV to expand the surveying area and provide effective input data for the bathymetry algorithms.

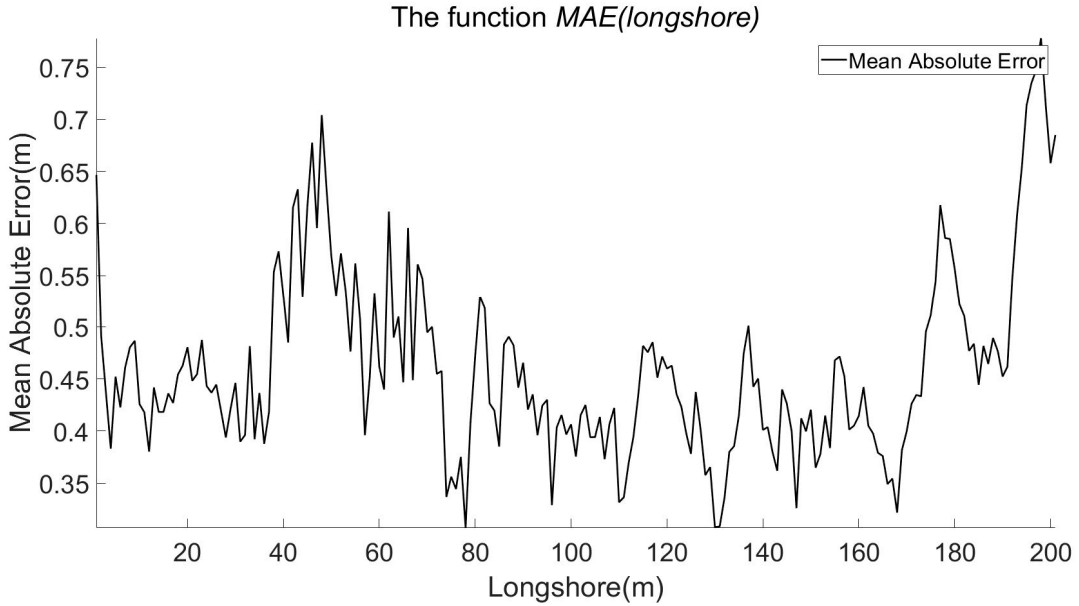

**Figure 20.** The diagram that MAE verses longshore.

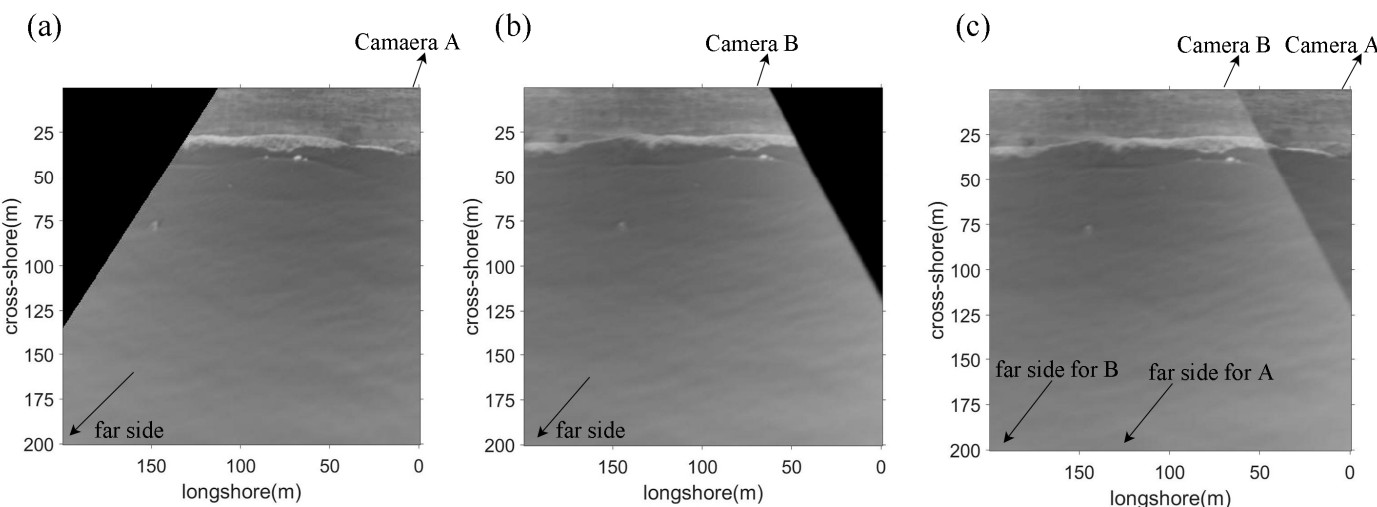

**Figure 21.** The far-side swell waves affect the results error of camera A alone, whereas camera B compensate for the rectification bias of camera A. (**a**) Orthoimage of camera A. (**b**) Orthoimage of camera B. (**c**) Orthoimage of panoramic image.

## 5. Discussion

### 5.1. Source of Error

Although the results in the previous section showed the present method can reasonably realize the bathymetric mapping algorithm based on video stitching, this method should be further discussed regarding the existing problems and its applicability. In addition, all of the given outcomes used a methodology that needed knowledge of GCPs to optimize the camera's exterior orientation. Improper operation of GCPs also introduces errors to experimental results, both in site layout and postprocessing.

Although the deviation of the beach affects the specific image position of the GCPs coordinates, when the frames were under orthographic rectification, the error of the pixel coordinates of each frame of GCPs would accumulate. Furthermore, in frames where GCPs are not visible, background identification plays a temporary role. Camera motion is estimated by identifying fixed feature points in the field, but the main fixed feature points were all distributed on the beach. Since it is impossible to place stationary structures on a sea surface that is filled with moving swell waves, a small mismeasurement of one-side feature points could lead to a significant bias in the far field. However, the bias, in this case, tended to be exposed only for long-term measurements. As shown in Figure 22, estimating camera motion using the image alone was equivalent to visual odometry [37], and the error accumulation was acceptable for a short-time bathymetric mapping. Otherwise, the monocular system will suffer from the scale-drift issue.

To obtain the final panoramic video, we transformed the input videos by the estimated stitching and stabilization variable in Equations (8)–(10). In the overlapping region (Figure 17), background on the shore was stitched satisfactorily, but not surprisingly, foreground objects such as swell waves had subtle ghost artifacts. As shown in Figure 23, we performed bathymetry inversion of the common field captured by the two UAVs separately, and unsurprisingly found that the two results were similar, whose RMSE was 0.2 m, MSE was 0.04 m$^2$ and MAE was 0.078 m. The main reason for this result was the slight parallax between the two cameras with similar camera Euler angles and relatively short distances, which can be summarized as parallax bias.

### 5.2. Wide Vertical FOV

An area worth discussing is the further cross-shore region, with dense information that loses a lot. Given the underestimation bias in this part of the region, it is still scarcely understood whether the method studied can solve this problem. Current experiments show that video stitching is usually based on the spatial domain. The base stitching method used

in this work is SIFT-based feature matching [32]. We attempted to add a UAV in the vertical direction (in front of the first camera, Figures 24–26) to capture distant fields, which tried to expand the length of the cross-shore. Nonetheless, in the process of feature matching, there are often mismatches due to the feature points being different from each other at different angles of view at every moment. Overall, a large parallax problem exists between cameras. When iterating with Equation (15), either the value $P_i(t)$ is divergent or the stitching results perform badly. In this case, video stitching probably worked when the cameras were close enough, but the vertical fields of the cross-shore did not meet our expectations (Figure 27).

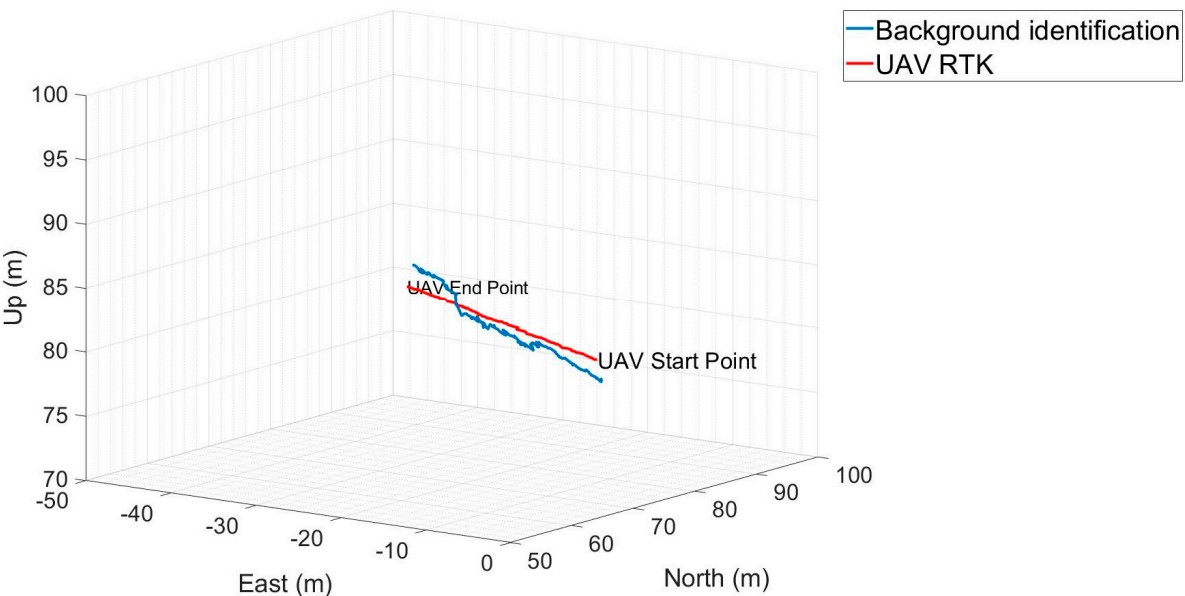

**Figure 22.** UAV RTK recorded the flight route, while background identification is to calculate the flight route.

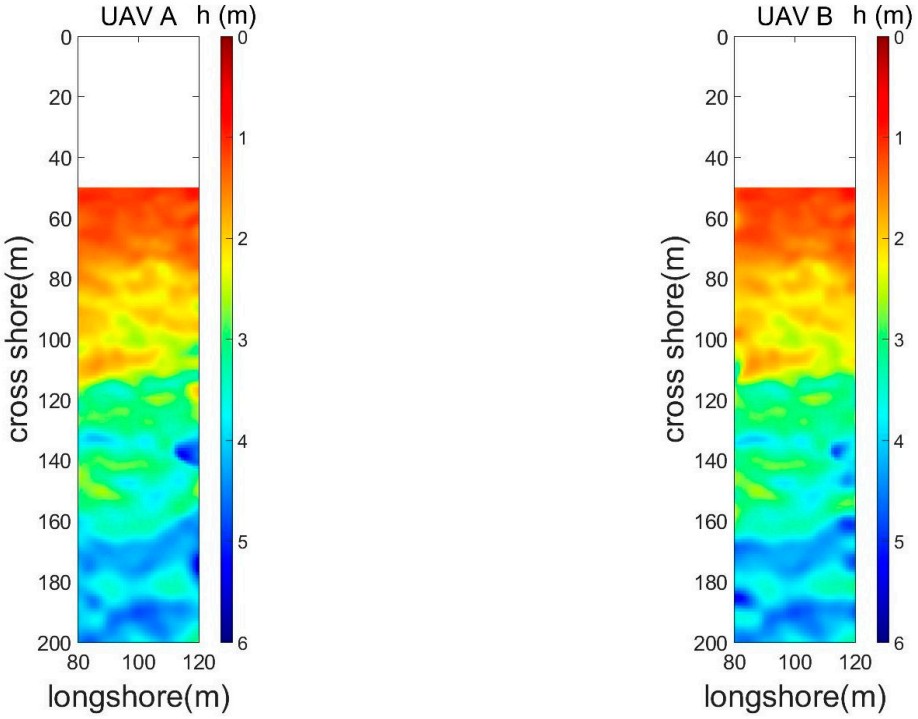

**Figure 23.** Bathymetry results of the common field captured by the two UAVs.

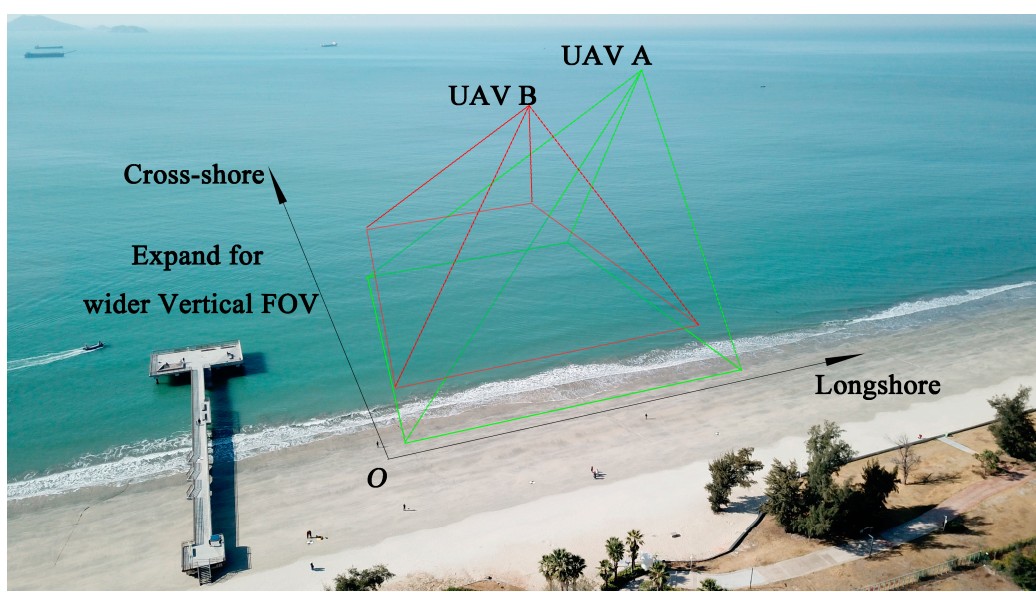

**Figure 24.** Two UAVs were placed to capture distant cross-shore.

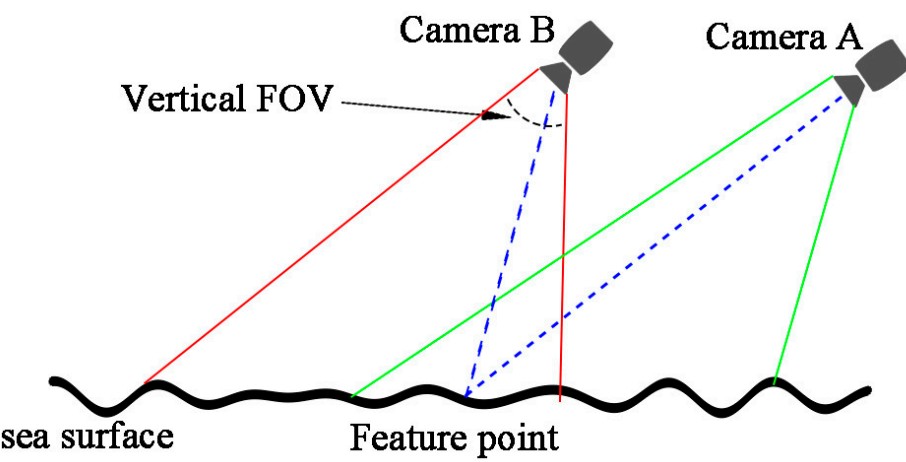

**Figure 25.** Inappropriate viewing angle cause strong parallax.

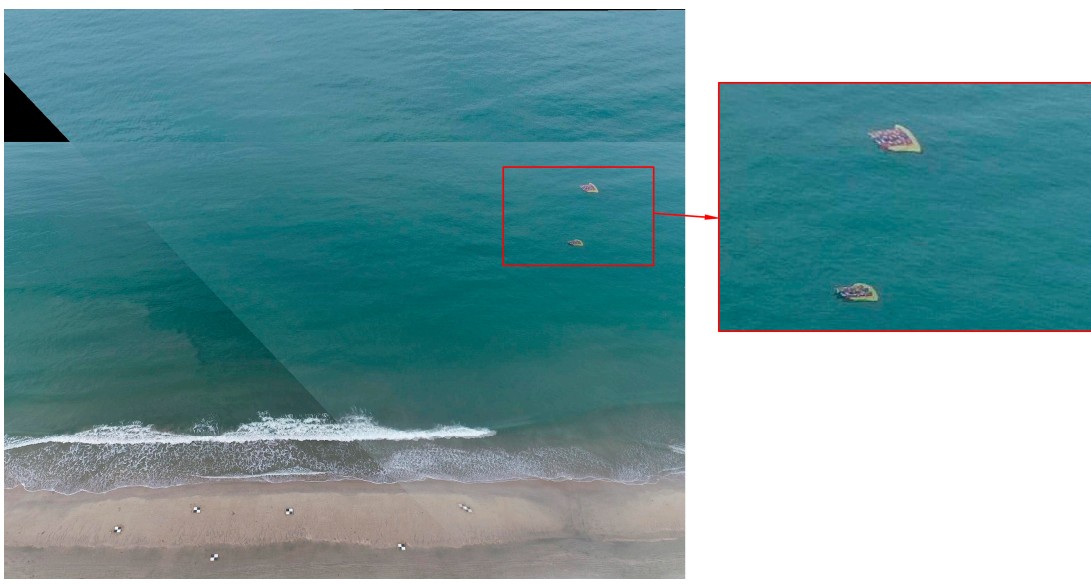

**Figure 26.** The result of a feature mismatch.

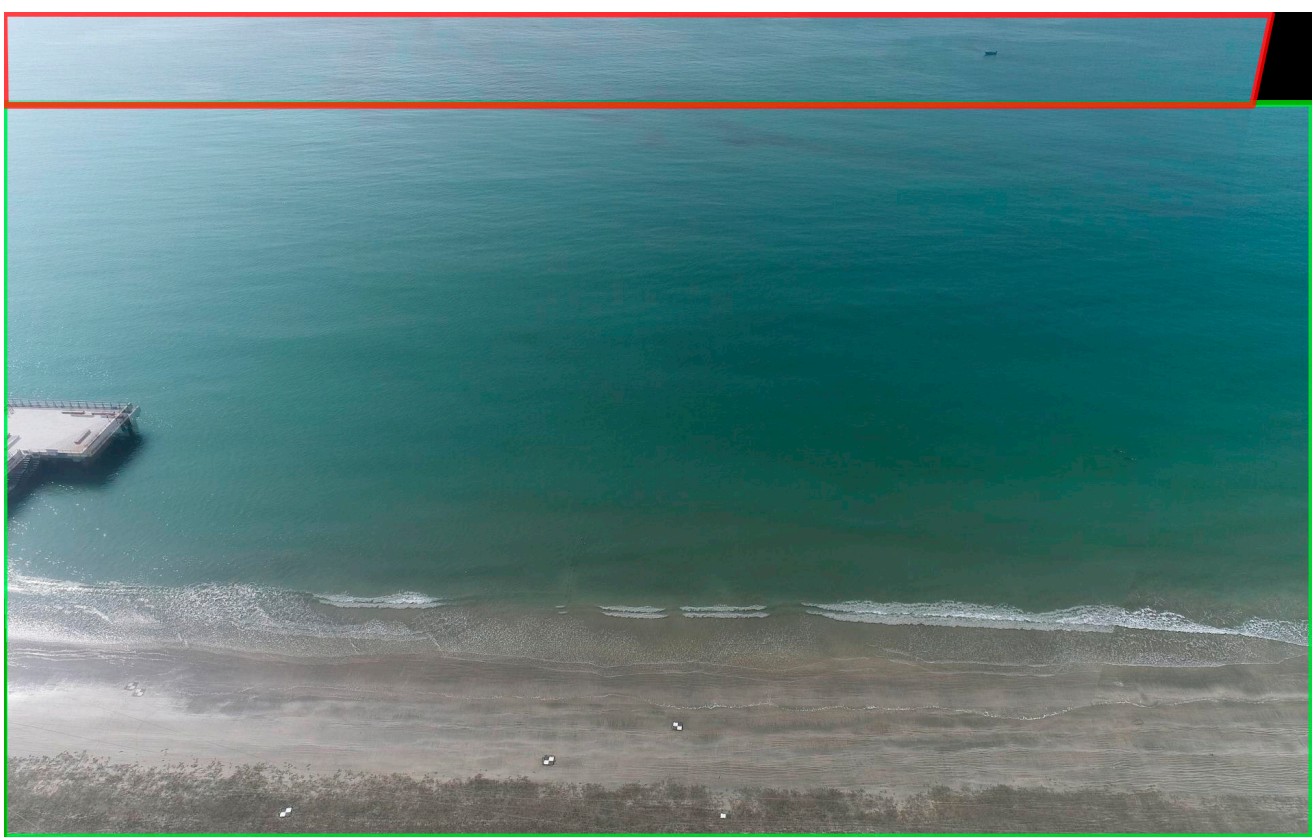

**Figure 27.** The green rectangle is the FOV of a single UAV. The red is the extension of the stitching.

Therefore, it is necessary to continue research related to the development of bathymetry algorithms based on optical cameras, using photogrammetric techniques based on UAV swarms. Taking into account the optical complexity of waters, one should put forward diverse ideas as well as practice experiments to verify theories.

## 6. Conclusions

This study developed a method of nearshore bathymetry surveying using UAVs' video stitching. By proving the effectiveness and limitations of single UAV bathymetry mapping, a second UAV was introduced to compensate for the shortcomings of the former. According to the results, this method expands the horizontal FOV of UAV mapping by planning the flight routes of two UAVs. Furthermore, while improving the efficiency of surveying, a part of the problem of how rectification bias affects the mapping results was solved. However, this work still cannot eliminate some errors for further cross-shore region, as well as parallax bias in stitching. This means that there is still work to be carried out on the research of the bathymetric mapping algorithm. Further research from the perspective of algorithm optimization will be carried out.

**Author Contributions:** Conceptualization, J.F. and H.P.; methodology, J.F.; software, J.F.; validation, J.F., H.P. and Z.L.; formal analysis, J.F.; investigation, J.F. and H.P.; resources, J.F., H.P. and Z.L.; data curation, J.F. and Z.L.; writing—original draft preparation, J.F.; writing—review and editing, J.F.; visualization, J.F.; supervision, J.F. and H.P.; project administration, J.F. All authors have read and agreed to the published version of the manuscript.

**Funding:** This research was supported in part by the Fundamental Research Funds for the Central Universities; and in part by Scientific Instruments Development Program of NSFC under Grant 61527810.

**Institutional Review Board Statement:** Not applicable.

**Informed Consent Statement:** Not applicable.

**Data Availability Statement:** The data that support the finding of this study are available on request from the corresponding author, H.P., upon reasonable request.

**Conflicts of Interest:** The authors declare no conflict of interest.

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
