# Peer review of "Surveying of Nearshore Bathymetry Using UAVs Video Stitching"

_jmse, doi:10.3390/jmse11040770_

Round 1

Reviewer 1 Report

Title & abstract:

The words ‘Video Stitching’ is quite vague for marine scientists and engineers, nonetheless used in the title (Surveying of Nearshore Bathymetry Using UAVs Video Stitching). Yet, the reader won’t be bothered as it shall be explained in the text where it would discover the meaning and the innovation behind. However, the abstract which shall inform swiftly the reader without the details, is not genuinely informative on the aforesaid ‘stitching’, i.e., merging of images, as it refers to the meaning and/or objective well indirectly when stating “video captured by a single UAV often shows a limited coastal zone with a lack of a wide field of view”: one understands that you (the authors) would stitch/join together multiple overlapping videos to generate a wide-FOV video. Notwithstanding this fuzziness on a topic which is not familiar to the marine scientists and engineers, i.e. computer vision & graphics, the relationship between the videos and bathymetry is not explicit until the reader stumbles on the last sentence of the abstract, with reference to the cBathy software which retrieves the bathymetry from the observation of wave motions: the videos are views of the sea-surface. Do all readers know cBathy and can infer that you pertain to waves? I doubt it. And readers could consider that the authors deal with the seafloor because they allude to “nearshore bathymetry”, and one can see the seafloor as well as the sea-surface in the nearshore (except when foam and bubbles mask it or in turbid waters). Nota: in the sentence “…cBathy and temporal cross-correlation analysis…”, one should invert the references to cbathy and timeCor (name of the temporal cross-correlation analysis) because timeCor, a method by Tuskada et al. 2020, is your main tool  to calculate depth.

I suggest reviewing the presentation of the case in the abstract,

-          highlighting the use of the technology (bathymetry estimation from the theory of small wave amplitude propagation, and related (linear) wave dispersion relationship, with wave celerity and frequency estimated from a sea-surface video obtained from UAVs;

-          then focusing on the impediments or shortcomings, last on the innovation(s) that mitigate the snags.

The  conclusions in §6 should be reflected in the abstract.

§1 Introduction:

I suggest to replace “More recently, some scholars have paid attention to obtaining the information they want from coastal video imagery based on the linear dispersion relationship…” (lines 50 & 51) by “More recently, some scholars have paid attention to obtaining the information they want from coastal video imagery  of waves based on the linear  dispersion relationship of waves’ propagation…”

In the equation (2), the variable f is not defined.

The figure 1 shows a wave with wavelength’s scale higher than the scale of the sea-floor variability, meaning that the equations (1) and (2) can’t apply as such because the waves can’t have a quasi linear dynamics as per the aforesaid equations in these conditions: the sea waves’ wavelength should be shorter, though we know that only long-period waves ensure accuracy of bathymetry estimation, but a filtered bathymetry. Nota: not only is it necessary to resolve shorter length scales found near beaches (e.g. sandbar structure) but to account for wave breaking, i.e. crest breaking -taking into account that video derived measurements of wavelength to apply the formula of linear small amplitude wave theory  is done between crests or areas of similar sun or sky glint signature (most of the time, when the seafloor gradients are low, waves do not plunge or collapse when breaking, hence “breaking” the original wave pattern and generating new waves, but spills, hence slowly dissipating its energy in the white water). It is only at line 109-110 that the reader is informed of the limitations of the methodology by “a suitable site needs to present a low-gradient slope in the intertidal area”. And only on line 114 that we should observe swells rather than wind waves.

Lines 79 to 81, which contain a statement whereby the current video taking scheme from UAVs is not efficient, do not provide an understandable argumentation, jumping to the intermediary conclusion, i.e. “simultaneous mapping of multiple cameras […] to obtain a wide field of view (FOV)”, then hopping to the conclusion that ‘it is not suitable if the cameras are in a single UAV’ at least commercial UAVs, reason why one should use multiple UAVs.

§2 Video processing:

One should give some explanations when using the words “intrinsic camera matrix” and “extrinsic camera matrix” at line 130 (the intrinsic parameters of a camera such as focal length, aperture, field-of-view, resolution, etc.; the extrinsic parameters of a camera depend on its location and orientation).

You inform of the brand and type of UAV, a DJI Phantom 4 RTK, but not of its version, e.g., the camera and lens (is it a fisheye?) which is now based on a 20Mpixel CMOS chip, of the RTK base station and the post-processing (accuracy of location), etc.

I suggest replacing “4K” at line 123 by “4K UHD TV” or “4K UHD” to help readers understand that it is four times the pixel count of full monitor HD displays (1920 x 1080 pixels) to which you have reduced the resolution of images. But they refer to this display under the usual video mode name of 1080p (writing it “1080P”), the relationship between these two standards being not evident to marine engineers: it would be better to replace “1080P” by “FHD”.

 Figure 3 is difficult to understand (with an insufficient title of the figure at line 165, i.e. “bundled paths” which refers to the concept of “bundled camera paths” of Liu et al., 2013) following the introduction of two techniques, i.e. “bundled video path and path optimization […] for video stitching” at lines 162-163 with no explanation to the layman: one would have expected a figure that would explain. Yet, the explanation of the figure is between line 166 and line 193 and targets the estimation of the camera path: one can then understand that “path optimization” means ‘estimation of the path of the camera, i.e. the time series of the camera location, by minimizing the errors of location of ground features vg’ -it is at least whet I’ve understood… (to understand it one should read at least 4 papers!). It is not conspicuous which ground features are used, because, at first glance it could not be a priori on the sea-surface because of its line structure, but Ground Control Points (GCPs) on the backshore -what about the grid meshes that are exclusively on water? It is only on figure 5.b that the reader realizes that on has GCPs at sea, probably foam packets (but edges or corners? and for what luminance discrepancy? It is only on page 19/line 482 that we learn that you used the SIFT method invented by Lowe in the early 2000s); and it is not clear what are the GCPs on the beach until we reach line 244 which informs that they are “black-and-white sheets” (implying that there was not much wind on-ground, and the pitch-roll-yaw of the UAVs by wind-shaking should have been very small, need of very small corrections by visual odometry). How are the feature points on land and at sea identified and selected? I was not able to find it in the text.

Nota: the legend of Figure 3 should refer to figure 4, or the two figures put together as 3.a and 3.b.

As for “stitching”, it is only introduced in an equation, equation 7, through a statement at line 186 “there is the following optimization function that achieves stitching and stabilization at the same time” when the word stabilization was only used once before, in the §2.3 title. To a layman like I am, videos stitching is when individual video footages are turned into a single video, usually for producing a single panorama video as if one shot with a larger FOV, but it pops up only at line 194. Equation 6 introduces 2 indices, i.e. A & B, but we understand only at line 190 that the indices refer to 2 cameras on board different UAVs, and that one optimizes simultaneously the estimation of the locations of the A-UAV and the B-UAV.

l and w of equation 5, as well as b of equations 7 and 9 are Lagrange multipliers, but the text does not inform on how they are obtained.

Equation 9 introduces to “the improvement of this paper” which would be detailed later in the paper but on a too light justification of “the above video path optimization process will change the camera’s optical center position and distortion degree to an extent, like as-projective-as-possible image warping in the method. Given the following orthorectification, the frames should be transformed as little as possible to maintain pixel information integrity”; but i. it does not give really the “extent” (sentence in English to be reviewed), ii. the word “orthorectification” pops up for the first time in the text (if knowing that orthorectification is the process of removing the effects of image perspective, the reader can understand, but if it is a satellite Earth Observation scientist, he/she would not understand because he/she will think of relief-terrain effect instead of tilt), and iii. why use the word “following” to justify a conclusion when it should be given before?  Because it is effectively in §2.4 downward that it is explained: calculation of an extrinsic camera matrix, but you use the same variable identifier M for a camera matrix and the camera motion path, which is confusing.

The variables R and T in equation 10 have not been declared previously and are found anew in equation 11 that helps the reader understand it refers to ‘rotation’ and ‘translation’ (if the word ‘rotation’ pops up at lines 241-242 above equ. 11, the word translation doesn’t). Half of the variables in equation 11 are not defined.

It is not understandable for a reader whether the step described in §2.4 predates the step of §2.3 or not, though it is not. While I understood  the need for an orthorectification, which, in my mind, could have been performed prior to the obtention of the optical flow, for a proper projection of the images on a plane  to get a proper location of pixels, I did not understand the statement “we hope to unify the pixel resolution in sequence images to facilitate the  implementation of a subsequent  algorithm” in §2.4 which was hammered anew in §3.1 “we solved the orthorectification of down-sampling images to ensure strong consistency in pixel resolution”: how is it done? a resampling?

§3 Signal extraction:

On line 274 “Figure 4” should be replaced by ‘Figure 7’?

 In §3.2 the word ‘tide’ in “however, many short wind waves unrelated to tide movement exist on the sea surface” means probably ‘swell’.

 The retrieved bathymetry on Figures 13 & 14 is not very good, with only one camera -yet probably good enough to calculate the mean beach slope which shall be the variable of interest (derivative of depth instead of depth itself). It would have been interesting to have some hints on the depth differences obtained on the 3 transects between timeCor and cBathy -otherwise why show the two results apart from claiming success from 3 experimental evidence (Table 3) which are statistically not sound enough? (all the more than the bathymetry timeCor-derived map of figure 12 is very noisy and the bathymetry timeCor-derived map of figure 13.a is much more noisy than the cBathy-derived map of figure 13.b). Providing clues from an analysis would help promote your timeCor method even if theoretical. It is simply stated that “the source of error mainly comes from the difference between each algorithm” and quite vague infos.

In §3.3 about the results on the stitched video:

To achieve dual UAV bathymetry, we adjusted the distribution of two UAVs” is not understandable in English, as a distribution is rather an arrangement or a mathematical function. Shall ‘distribution’ not replaced by ‘placements’?

Correspondingly, we calculated the global Mean Square Error (MSE) and the global RMSE, whose values were 0.37m² and 0.60m respectively”: for which derived depths? the cBathy one or the timeCor ones? If these figures are important to draw the following “It can be concluded that our method is valid and meets expectations”, references of the objectives shall be supplied, or, at least the MSE of the §3.2 cases as a comparison.

§4 Bathymetry results:

What is the significance of figure 18 with a calculation of the MAE (discrepancy between results of timeCor and cBathy)? It is expected to have an analysis of the differences in addition to an assessment of their scale.

We would foresee that the bathymetry results in areas where the videos of the two UAVs overlap would be better than when one calculates the bathymetry. Figure 20 shows bathymetry derived from videos taken by the sensors on the two UAVs which are processed independently: if the case, what’s the sense of stitching the videos, the main topic of this paper? I believe this figure is just there to illustrate that the stitching does not affect the accuracy of the derived bathymetry measurements when I had hoped to improve it as I can see waves crossing the borders between videos. It is only in §5.2 that we’re informed that “in the overlapping region [between the videos] foreground objects like swell waves have subtle ghost artifacts” which might prevent using the stitched video to calculate depth in the overlapping areas? or might create errors?

§5 Discussion:

Most of my questions, or questions of the readers when being acquainted with the results of the paper, should get answers in the §5 Discussion.

Yet, §5.1 about ‘source of error’ (without a s to ‘source’ and ‘error’) do not refer to an error or errors that would have been denounced in the previous §s, but to errors that one can be seen effectively on figures and were not mentioned before;  nevertheless it is somewhat identical in content to §5.1 of the paper by Tuskada et al. 2020 from which stems the timeCor method you employed . If, adding to Tuskada et al.’s statements, you introduced a potential method of mixed visual odometry & GNSS -RTK recently published (Cao et al., 2021 -but published in 2022) to instil originality, it is not applied anyhow.

I did not understand the paragraph from line 449 to 452 about the origin of the “rectification bias” (information compression?) and the mitigation (compensation by interpolation?). The following lines (459 to 475) do not clarify it, as we just get a demonstration that there is a “parallax bias” between the results of the bathymetry estimation with one camera vs. the other.

§5.2: it explains how the UAVs were formed-up, i.e. relatively placed, to “to capture distant fields, […] to expand the length of the cross-shore”. However none of the figures show that the video stitching did it; we only realize that there was effectively an expansion of the FOV which could have been obtained with 2 cameras on the same UAV. The text should be reviewed to make sure that there is no confusion as the one that I call the attention to.

The text refers to “a large parallax problem [which] exists between cameras” to justify a bad performance of the stitching process, when it's mentioned  “a slight parallax between the two cameras” in the previous paragraph: these coexisting statements are upsetting . And we were not informed before of any bad completion of the stitching.

Nota: the statement “in this work, nearshore bathymetry within a wide horizontal FOV could be obtained with sub-meter accuracy” is interesting as far as people are interested in the depth as such, i.e. for maritime/ shipping security, access to ports, …, but it is mostly the slope which matters for coastal engineers ? If the case, could we get results accordingly?

§6 Conclusion:

The conclusion is not self-explanatory even for a reader who has read carefully the text development (from §1 to 5).

·         E.g., “according to the results, this method expands the horizontal FOV of UAV mapping by adjusting the distance of UAVs and Euler angles”: there is no development in the text about the adjustment, i.e. the optimization of the inter-distance of the 2 UAVs of the orientations of the cameras (taking into account that the terms “Euler angles” were introduced lately in §5.1 at lines 464 and 474, but not defined).

·         E.g., “while improving the efficiency of surveying and mapping, a part of the problem of how rectification bias affects the mapping results has been solved” means that we would now have the knowledge / understanding of the relationship between ‘rectification biases’ and ‘mapping results’: ‘mapping’ means ‘location of the bathymetry estimate’, or ‘bathymetry estimate’?

How shall the current work with 2 UAVs expand to work with “UAV swarms” in the “perspective of large-scale bathymetric mapping […] feasibility”?  There is no description of the optimization of UAVs’ deployment, paths, and cameras’ Euler angles in the current case of two UAVs. I suggest to write a development in the core of this paper  so as to help draw the roadmap with swarms of UAVs

Nota: uncommonly, the conclusion is not reflected in the abstract.

Remarks:

this paper deals with 3 topics, i.e., i. the use of UAVs, alone, or in form-up, ii. snapshots/videos, image processing, and mapping, iii. bathymetry retrieval. Most of the readers do not have the expertise simultaneously in the 3 fields; consequently, it is critical to deliver a text which reminds of the basics in the related engineering disciplines, yet jumping to results/methodologies already published. If the references to the papers are OK, it's not the case for the basics. I suggest reviewing the text accordingly, and even restructure according to the logic 'objectives (bathymetry retrieval & sea-floor mapping) - 'UAV use' - 'video shooting& processing' -results for mapping, but in three steps: a. background,  b. authors' innovation incl. new methods and results, and c. roadmap. It would facilitate the undertaking of the conclusion.  

Author Response

Thank you for taking the time to comment on my paper, I have made major revisions to the whole paper according to your request. I have commented on all the suggestions you mentioned.  Due to time constraints, there are still some pictures and citations that have not had time to be typeset. Thanks again for your review.

Reviewer 2 Report

The research is pretty interesting inspite of the method is not new. This study, ingeneral, gives some interesting information about various methods to retrieve coastal bathymetry data from UAVs.

However, the current manuscript is not written well,  mostly the Abtract, Introduction and Conclusion Sections. For example, in the Introduction Part, there are several conclusions come without their sources, and seem like the they come from the authors's understanding. If so, please provide the sources, and cite in the manuscript. More details, please see in the file attached.

Regarding the technique applied here, I did not see the effect of tide. In most of the coastal area, tidal effects play an important factor in the accuracy of the water depth derived from space photos. Please mention its effects in the revised manuscript.

Round 2

Reviewer 2 Report

Thank you very much for your hard work. I understand that the authors have significantly revised the manuscript. 

I am satisfied with the current form of the revised version.